# Scalable Multi-Output Gaussian Processes with Stochastic Variational Inference

**Xiaoyu Jiang**               *xiaoyu.jiang@postgrad.manchester.ac.uk*
*Department of Computer Science*
*University of Manchester, Manchester*

**Sokratia Georgaka**             *sokratia.georgaka@manchester.ac.uk*
*Faculty of Biology, Medicine and Health*
*University of Manchester, Manchester*

**Magnus Rattray**              *magnus.rattray@manchester.ac.uk*
*Faculty of Biology, Medicine and Health*
*University of Manchester, Manchester*

**Mauricio A. Álvarez**           *mauricio.alvarezlopez@manchester.ac.uk*
*Department of Computer Science*
*University of Manchester, Manchester*

**Reviewed on OpenReview:** *https://openreview.net/forum?id=kKOWrBZALi*

## Abstract

The Multi-Output Gaussian Process (MOGP) is a popular tool for modelling data from multiple sources. A typical choice to build a covariance function for an MOGP is the Linear Model of Coregionalisation (LMC), which parametrically models the covariance between outputs. The Latent Variable MOGP (LV-MOGP) generalises this idea by modelling the covariance between outputs using a kernel applied to latent variables, one per output, leading to a flexible MOGP model that allows efficient generalisation to new outputs with few data points. The computational complexity in LV-MOGP grows linearly with the number of outputs, which makes it unsuitable for problems with a large number of outputs. In this paper, we propose a stochastic variational inference approach for the LV-MOGP that allows mini-batches for both inputs and outputs, making computational complexity per training iteration independent of the number of outputs. We demonstrate the performance of the model by benchmarking against some other MOGP models in several real-world datasets, including spatial-temporal climate modelling and spatial transcriptomics.

## 1 Introduction

Gaussian processes (GP) have established themselves as a powerful and flexible tool for modelling nonlinear functions within a non-parametric Bayesian framework (Williams & Rasmussen, 2006). Multi-output Gaussian processes (MOGP) generalise this powerful framework to the vector-valued random field (Alvarez et al., 2012) by capturing correlations not only across different inputs but also across different output functions. This characteristic has been experimentally shown to provide better predictions in fields such as geostatistics (Wackernagel, 2003), heterogeneous regression (Moreno-Muñoz et al., 2018), and modelling of aggregated (Yousefi et al., 2019) and hierarchical datasets (Ma et al., 2023).

The primary focus in the literature on MOGP has been on developing an appropriate cross-covariance function between multiple outputs. Two classical approaches for defining such cross-covariance functions are the Linear Model of Coregionalisation (LMC) (Journel & Huijbregts, 1976; Goovaerts, 1997) and process convolutions (Higdon, 2002). In the former, each output corresponds to a weighted sum of shared latent random functions.

In the latter, each output is modelled as the convolution integral between a smoothing kernel and a latent random function common to all outputs. The Latent Variable MOGP (LV-MOGP), introduced by Dai et al. (2017), extends the construction of the output covariance by applying a kernel function to latent variables, one for each output. This approach enables efficient generalisation to new outputs. Dai et al. (2017) also demonstrated experimentally that LV-MOGP outperforms LMC, which tends to face overfitting issues when estimating a full-rank coregionalisation matrix.

To address the cubic complexity concerning the number of outputs in MOGP (Bonilla et al., 2007; Alvarez et al., 2012), Nguyen et al. (2014) proposes to use mini-batches in the context of the LMC framework, making the computational complexity for each iteration independent of the size of the outputs. However, the model's parameters increase linearly with the number of outputs, constraining its practical scalability to problems with large-scale output. The computational complexity associated with estimating the marginal likelihood for the LV-MOGP also increases linearly with the number of outputs, making it unsuitable for problems with a large number of outputs. The stochastic formulation allowing minibatch training of Bayesian Gaussian Process Latent Variables Models (BGPLVM) has been investigated in Lalchand et al. (2022), employing stochastic variational inference (SVI) (Hoffman et al., 2013; Hensman et al., 2013) to facilitate scalable inference. However, BGPLVMs are applied in unsupervised learning contexts, which distinguishes them from the MOGP models considered in supervised learning settings.

In this paper, we adapt the SVI techniques used in BGPLVM to LV-MOGP to formulate a training objective that supports mini-batching for both inputs and outputs. This approach makes the computational complexity per training iteration independent of the number of outputs. This enhances the accessibility of our models for problems with a large number of outputs ($\geq$5000), an area that has been less explored by previous MOGP methods. Furthermore, we generalise the assumption of latent variables in LV-MOGP by introducing multiple latent variables for each output, allowing the construction of more flexible covariances. Our *doubly stochastic* training objective decomposes the data-dependent term across data points, allowing trivial marginalisation of *missing values*. Moreover, our framework easily extends to non-Gaussian likelihoods, making our model applicable to a wide range of datasets, such as modelling count data using a Poisson likelihood. We refer to our approach as the Generalised Scalable Latent Variable MOGP (GS-LVMOGP). We test our model on several real-world data sets, such as spatiotemporal temperature modelling and spatial transcriptomics.

## 2 Background

For multi-task modelling of a dataset collected from $D$ sources with inputs $\mathbf{X} = \{\mathbf{x}_n \in \mathbb{R}^{Q_X}\}_{n=1}^N$ and observations $\mathbf{Y} = \{\mathbf{y}_d\}_{d=1}^D$, where $\mathbf{y}_d = \{y_{dn}\}_{n=1}^N$, multiple output Gaussian processes (MOGPs) induce a prior distribution over vector-valued functions by ensuring any finite collection of function values $f_{d_1}(\mathbf{x}_1), f_{d_2}(\mathbf{x}_2), ..., f_{d_n}(\mathbf{x}_n)$ with $(d_i)_{i=1}^n \subseteq \{1, 2, ..., D\}$ are multivariate Gaussian distributed. The Linear Model of Coregionalisation (LMC) and Latent Variable MOGP (LV-MOGP) are two approaches used to define such a prior.

**The Linear Model of Coregionalisation** In the LMC framework, every output (source) is modelled as a *linear* combination of independent random functions (Journel & Huijbregts, 1976). If the independent random functions are Gaussian processes, then the resulting model will also be a Gaussian process (Alvarez & Lawrence, 2011). For output $d$, the model is expressed as: $f_d(\mathbf{x}) = \sum_{q=1}^Q \sum_{i=1}^{R_q} a_{d,q}^i u_q^i(\mathbf{x})$, where the functions $\{u_q^i(\mathbf{x})\}_{i=1}^{R_q}$ are latent Gaussian processes sharing the same covariance function $k_q(\mathbf{x}, \mathbf{x}')$. There are $Q$ groups of functions, with each member of a group sharing the same kernel function, but sampled independently. The cross-covariance between any two functions $f_d(\mathbf{x})$ and $f_{d'}(\mathbf{x}')$ at inputs $\mathbf{x}$ and $\mathbf{x}'$ is given by: $\text{cov}\left[f_d(\mathbf{x}), f_{d'}(\mathbf{x}')\right] = \sum_{q=1}^Q \sum_{i=1}^{R_q} a_{d,q}^i a_{d',q}^i k_q(\mathbf{x}, \mathbf{x}') = \sum_{q=1}^Q b_{d,d'}^q k_q(\mathbf{x}, \mathbf{x}')$, with $b_{d,d'}^q = \sum_{i=1}^{R_q} a_{d,q}^i a_{d',q}^i$. For $N$ inputs, we denote the vector of values from the output $d$ evaluated at $\mathbf{X}$ as $\mathbf{f}_d$. The stacked version of all outputs is defined as $\mathbf{f}$, so that $\mathbf{f} = [\mathbf{f}_1^\top, ..., \mathbf{f}_D^\top]^\top$. Now the covariance matrix for the joint process over $\mathbf{f}$ is expressed as:

$$\mathbf{K}_{\mathbf{f},\mathbf{f}} = \sum_{q=1}^Q \mathbf{A}_q \mathbf{A}_q^\top \otimes \mathbf{K}_q = \sum_{q=1}^Q \mathbf{B}_q \otimes \mathbf{K}_q, \tag{1}$$

where the symbol $\otimes$ denotes the Kronecker product, $\mathbf{A}_q \in \mathbb{R}^{D \times R_q}$ has entries $a_{d,q}^i$ and $\mathbf{B}_q \in \mathbb{R}^{D \times D}$ has entries $b_{d,d'}^q$ and is known as the *coregionalisation matrix*.

As a simplified version of the LMC, the intrinsic coregionalisation model (ICM) assumes that the elements $b_{d,d'}^q$ of the coregionalisation matrix $\mathbf{B}_q$ can be written as $b_{d,d'}^q = v_{d,d'} b_q$. This simplifies the model, making the intrinsic coregionalisation model a linear model of coregionalisation with $Q = 1$ (Alvarez & Lawrence, 2011). In this case, Eq. 1 can be expressed as $\mathbf{K}_{\mathbf{f},\mathbf{f}} = \mathbf{A}_c \mathbf{A}_c^\top \otimes \mathbf{K}_c = \mathbf{B}_c \otimes \mathbf{K}_c$.

**Latent Variable MOGP**   In ICM, the coregionalisation matrix $\mathbf{B}_c$ is directly parametrised by its matrix factor $\mathbf{A}_c$. Latent Variable MOGP (LV-MOGP) (Dai et al., 2017) tried an alternative approach, that is, constructing coregionalisation matrices $\mathbf{B}_c$ using a kernel applied to latent variables, one per output. Denoting the latent variables as $\mathbf{H} = \{\mathbf{h}_d\}_{d=1}^D$, where $\mathbf{h}_d \in \mathbb{R}^{Q_H}$ is the latent variable assigned to output $d$. The covariance between outputs is then computed as $\mathbf{K}^H = k^H(\mathbf{H}, \mathbf{H})$, where $k^H$ is the kernel defined on latent variable space. The covariance between inputs $\mathbf{X}$ is captured by $\mathbf{K}^X = k^X(\mathbf{X}, \mathbf{X})$, where $k^X$ is another kernel defined on the input space. The covariance matrix $\mathbf{K}_{\mathbf{f},\mathbf{f}}$ is defined as: $\mathbf{K}_{\mathbf{f},\mathbf{f}} = \mathbf{K}^H \otimes \mathbf{K}^X$. Latent variables $\mathbf{H}$ are treated in a Bayesian manner, with prior distribution $\mathbf{h}_d \sim \mathcal{N}(\mathbf{h}_d \mid \mathbf{0}, \mathbb{I}_{Q_H})$. The probabilistic distributions of LV-MOGP are defined as:

$$p(\mathbf{h}_d) = \mathcal{N}(\mathbf{h}_d \mid \mathbf{0}, \mathbb{I}_{Q_H}), \quad p(\mathbf{f} \mid \mathbf{X}, \mathbf{H}) = \mathcal{N}(\mathbf{f} \mid \mathbf{0}, \mathbf{K}_{\mathbf{f},\mathbf{f}}), \quad p(\mathbf{Y} \mid \mathbf{f}) = \mathcal{N}(\mathbf{Y} \mid \mathbf{f}, \sigma^2 \mathbb{I}_{ND}), \tag{2}$$

where $\sigma$ is the Gaussian likelihood parameter.

**Variational Inference**   To perform posterior inference in LV-MOGP defined in Eq. 2, Dai et al. (2017) derive a variational lower bound using auxiliary variables (Titsias, 2009; Titsias & Lawrence, 2010; Hensman et al., 2013). They place inducing points in both input space and latent space, which are denoted as $\mathbf{Z}^X = \{\mathbf{z}_1^X, \mathbf{z}_2^X, ..., \mathbf{z}_{M_X}^X\}$ and $\mathbf{Z}^H = \{\mathbf{z}_1^H, \mathbf{z}_2^H, ..., \mathbf{z}_{M_H}^H\}$ respectively. The inducing variables $\mathbf{u}$ follows the same Gaussian process prior:

$$\mathcal{N}(\mathbf{u} \mid \mathbf{0}, \mathbf{K}_{\mathbf{u},\mathbf{u}}) = \mathcal{N}(\mathbf{u} \mid \mathbf{0}, \mathbf{K}_{\mathbf{u},\mathbf{u}}^H \otimes \mathbf{K}_{\mathbf{u},\mathbf{u}}^X), \tag{3}$$

where $\mathbf{K}_{\mathbf{u},\mathbf{u}}^H = k^H(\mathbf{Z}^H, \mathbf{Z}^H)$, and $\mathbf{K}_{\mathbf{u},\mathbf{u}}^X = k^X(\mathbf{Z}^X, \mathbf{Z}^X)$. The conditional distribution of $\mathbf{f}$ given $\mathbf{u}$ is:

$$p(\mathbf{f} \mid \mathbf{u}, \mathbf{Z}^H, \mathbf{Z}^X, \mathbf{H}, \mathbf{X}) = \mathcal{N}\Big(\mathbf{f} \mid \mathbf{K}_{\mathbf{f},\mathbf{u}} \mathbf{K}_{\mathbf{u},\mathbf{u}}^{-1} \mathbf{u}, \mathbf{K}_{\mathbf{f},\mathbf{f}} - \mathbf{K}_{\mathbf{f},\mathbf{u}} \mathbf{K}_{\mathbf{u},\mathbf{u}}^{-1} \mathbf{K}_{\mathbf{u},\mathbf{f}}\Big), \tag{4}$$

where $\mathbf{K}_{\mathbf{f},\mathbf{u}} = \mathbf{K}_{\mathbf{f},\mathbf{u}}^H \otimes \mathbf{K}_{\mathbf{f},\mathbf{u}}^X$ and $\mathbf{K}_{\mathbf{f},\mathbf{u}}^H = k^H(\mathbf{H}, \mathbf{Z}^H)$, and $\mathbf{K}_{\mathbf{f},\mathbf{u}}^X = k^X(\mathbf{X}, \mathbf{Z}^X)$. They approximate the posterior distribution $p(\mathbf{f}, \mathbf{u}, \mathbf{H} \mid \mathbf{Y})$ by $p(\mathbf{f} \mid \mathbf{u}, \mathbf{H})q(\mathbf{u})q(\mathbf{H})$, where $q(\mathbf{u}) = \mathcal{N}(\mathbf{u} \mid \mathbf{M}^{\mathbf{u}}, \boldsymbol{\Sigma}^{\mathbf{u}})$, and $q(\mathbf{H}) = \prod_{d=1}^D \mathcal{N}(\mathbf{h}_d \mid \mathbf{M}_d, \boldsymbol{\Sigma}_d)$, where $\mathbf{M}^{\mathbf{u}}, \boldsymbol{\Sigma}^{\mathbf{u}}, \{\mathbf{M}_d, \boldsymbol{\Sigma}_d\}_{d=1}^D$ are parameters to be estimated. The evidence lower bound (ELBO) can be derived as:

$$\log p(\mathbf{Y} \mid \mathbf{X}) \geq \underbrace{\mathbb{E}_{p(\mathbf{f} \mid \mathbf{u}, \mathbf{X}, \mathbf{H})q(\mathbf{u})q(\mathbf{H})}[\log p(\mathbf{Y} \mid \mathbf{f})]}_{\mathcal{F}} - \text{KL}(q(\mathbf{u})||p(\mathbf{u})) - \text{KL}(q(\mathbf{H})||p(\mathbf{H})),$$

where $\mathcal{F}$ has a closed-form solution (Dai et al., 2017), see Appendix A.1:

$$\mathcal{F} = -\frac{ND}{2}\log 2\pi\sigma^2 - \frac{1}{2\sigma^2}\mathbf{Y}^\top\mathbf{Y} - \frac{1}{2\sigma^2}\text{Tr}(\mathbf{K}_{\mathbf{u},\mathbf{u}}^{-1}\Phi\mathbf{K}_{\mathbf{u},\mathbf{u}}^{-1}(\mathbf{M}^{\mathbf{u}}(\mathbf{M}^{\mathbf{u}})^\top + \boldsymbol{\Sigma}^{\mathbf{u}})) + \frac{1}{\sigma^2}\mathbf{Y}^\top\Psi\mathbf{K}_{\mathbf{u},\mathbf{u}}^{-1}\mathbf{M}^{\mathbf{u}}$$

$$- \frac{1}{2\sigma^2}(\psi - \text{Tr}(\mathbf{K}_{\mathbf{u},\mathbf{u}}^{-1}\Phi)),$$

where $\psi = \langle \text{Tr}(\mathbf{K}_{\mathbf{f},\mathbf{f}}) \rangle_{q(\mathbf{H})}$, $\Psi = \langle \mathbf{K}_{\mathbf{f},\mathbf{u}} \rangle_{q(\mathbf{H})}$ and $\Phi = \langle \mathbf{K}_{\mathbf{u},\mathbf{f}}\mathbf{K}_{\mathbf{f},\mathbf{u}} \rangle_{q(\mathbf{H})}$. Notice that the computational complexity of the $\mathcal{F}$ term increases linearly with both $D$ and $N$, [1] rendering the method unsuitable for applications involving a large number of outputs and inputs.

---

[1]This is true for terms $\mathbf{Y}^\top\mathbf{Y}$, $\psi$, $\Psi$ and $\Phi$.

**Connections between LMC and LV-MOGP**  The relationship between these two approaches can be established by interpreting the rows of the matrix factors $\mathbf{A}_q$ of coregionalisation matrices in LMC as $R_q$-dimensional latent variables. In this context, the coregionalisation matrices $\mathbf{B}_q$ take the form of kernel matrices, with a linear kernel function applied to these latent variables. When $Q = 1$, LV-MOGP can be seen as an extension of LMC, achieved by replacing the linear kernel function with any valid kernel function. However, unlike LMC, which directly optimises $\mathbf{A}_q$, LV-MOGP introduces variational distributions $q(\mathbf{H})$ for these latent variables, thereby "variationally integrating" them during the optimisation process. This distinction helps mitigate the risk of overfitting, as noted in (Titsias & Lawrence, 2010). This connection also motivates the extension of LV-MOGP to $Q > 1$, resulting in a model that incorporates a sum of separable kernels, similar to LMC.

## 3 Generalised Scalable LV-MOGP

We now investigate the stochastic formulation of LV-MOGP and extend its assumption regarding latent variables. Instead of a single latent variable per output in LV-MOGP, we propose the use of possibly $Q \geq 1$ latent variables, denoted as $\mathbf{H}_d = \{\mathbf{h}_{d,1}, \mathbf{h}_{d,2}, ..., \mathbf{h}_{d,Q}\}$, for $d \in \{1, 2, ..., D\}$.

For simplicity, we assume all these latent variables have the same dimensionality $Q_H$ and are independent of each other. The prior distribution of the latent variables is then defined as follows:

$$p(\mathbf{H}) = \prod_{d=1}^{D} p(\mathbf{H}_d) = \prod_{d=1}^{D} \prod_{q=1}^{Q} p(\mathbf{h}_{d,q}), \tag{5}$$

where $p(\mathbf{h}_{d,q}) = \mathcal{N}(\mathbf{h}_{d,q} \mid \mathbf{0}, \mathbb{I}_{Q_H})$ if there is no extra information. We may have additional information about the meaning of these latent variables and therefore for particular applications, we can use different mean vectors or covariances per latent variable, such as in the spatiotemporal dataset in the experimental section 5, we assume the prior mean vectors correspond to the initial location of each output. The generalised LV-MOGP model is defined as:

$$p(\mathbf{f} \mid \mathbf{H}, \mathbf{X}) = \mathcal{N}(\mathbf{f} \mid \mathbf{0}, \underbrace{\sum_{q=1}^{Q} \mathbf{K}_q^H \otimes \mathbf{K}_q^X}_{\mathbf{K}_{\mathbf{f},\mathbf{f}}}); \quad p(\mathbf{y}_d \mid \mathbf{f}_d) = \mathcal{N}(\mathbf{y}_d \mid \mathbf{f}_d, \sigma_d^2 \mathbb{I}_N), \tag{6}$$

where $\mathbf{K}_q^H$ represents the covariance matrix computed on $\mathbf{H}_q = \{\mathbf{h}_{1,q}, \mathbf{h}_{2,q}, ..., \mathbf{h}_{D,q}\}$ using the $q$-th kernel function on the latent space, denoted as $k_q^H$. Similarly, $\mathbf{K}_q^X$ represents the covariance matrix computed on $\mathbf{X}$ with $q$-th kernel function on the input space, denoted as $k_q^X$. $\sigma_d$ is the likelihood parameter for output $d$. When $Q = 1$, the model is reduced to LV-MOGP; however, when $Q > 1$, our model provides greater flexibility for constructing the covariance matrix.

### 3.1 Auxiliary Variables

We employ auxiliary variables (Titsias, 2009; Hensman et al., 2013) to facilitate efficient learning and inference. Similar to LV-MOGP, we consider inducing locations in both input and latent spaces. We assume $M_X$ inducing inputs in input space, denoted as $\mathbf{Z}^X = \{\mathbf{z}_1^X, \mathbf{z}_2^X, ..., \mathbf{z}_{M_X}^X\}$, where $\mathbf{z}_i^X \in \mathbb{R}^{Q_X}, \forall i \in \{1, 2, ..., M_X\}$. Distinct from LV-MOGP, the inducing locations in latent space are composed of $Q$ components. There are $M_H$ inducing locations in latent space, with the $i$-th inducing latent location being $\mathbf{Z}_i^H = \{\mathbf{z}_{i,1}^H, \mathbf{z}_{i,2}^H, ..., \mathbf{z}_{i,Q}^H\}$, and each component $\mathbf{z}_{i,q}^H \in \mathbb{R}^{Q_H}$. The $M_H$ inducing locations are collectively denoted as $\mathbf{Z}^H = \{\mathbf{Z}_1^H, \mathbf{Z}_2^H, ..., \mathbf{Z}_{M_H}^H\}$. The inducing variables $\mathbf{u}$ follow the prior distribution

$$p(\mathbf{u} \mid \mathbf{Z}^H, \mathbf{Z}^X) = \mathcal{N}(\mathbf{u} \mid \mathbf{0}, \underbrace{\sum_{q=1}^{Q} \mathbf{K}_{\mathbf{u},\mathbf{u};q}^H \otimes \mathbf{K}_{\mathbf{u},\mathbf{u};q}^X}_{\mathbf{K}_{\mathbf{u},\mathbf{u}}}), \tag{7}$$

where $\mathbf{K}_{\mathbf{u},\mathbf{u};q}^H$ is the covariance matrix computed on $\mathbf{Z}_q^H = \{\mathbf{z}_{1,q}^H, \mathbf{z}_{2,q}^H, ..., \mathbf{z}_{M_H,q}^H\}$ with kernel function $k_q^H$ and $\mathbf{K}_{\mathbf{u},\mathbf{u};q}^X$ is the covariance matrix on $\mathbf{Z}^X$ with kernel function $k_q^X$. The conditional distribution of $\mathbf{f}$ given inducing variables $\mathbf{u}$ follows as Eq. 4 where $\mathbf{K}_{\mathbf{f},\mathbf{u}} = \sum_{q=1}^Q \mathbf{K}_{\mathbf{f},\mathbf{u};q}^H \otimes \mathbf{K}_{\mathbf{f},\mathbf{u};q}^X$, $\mathbf{K}_{\mathbf{f},\mathbf{u};q}^H = k_q^H(\mathbf{H}_q, \mathbf{Z}_q^H)$ and $\mathbf{K}_{\mathbf{f},\mathbf{u};q}^X = k_q^X(\mathbf{X}, \mathbf{Z}_q^H)$.

## 3.2 Variational Distributions

The log marginal likelihood is not tractable due to the presence of latent variables. Therefore, we use variational inference to compute a lower bound of the original log marginal likelihood. Specially, we employ *mean field* variational distribution for latent variables $\mathbf{H}$, i.e.

$$q(\mathbf{H}) = \prod_{d=1}^D q(\mathbf{H}_d) = \prod_{d=1}^D \prod_{q=1}^Q q(\mathbf{h}_{d,q}), \tag{8}$$

where $q(\mathbf{h}_{d,q}) = \mathcal{N}(\mathbf{h}_{d,q} \mid \mathbf{m}_{d,q}, \mathrm{Diag}(\mathbf{S}_{d,q}))$, $\mathbf{m}_{d,q}, \mathbf{S}_{d,q} \in \mathbb{R}^{Q_H}$ and $\mathrm{Diag}(\mathbf{S}_{d,q})$ denotes the construction of a diagonal matrix with the elements of $\mathbf{S}_{d,q}$ placed on the diagonal. For inducing variables $\mathbf{u}$, the variational distribution is: $q(\mathbf{u}) = q(\mathbf{u} \mid \mathbf{M}_{\mathbf{u}}, \boldsymbol{\Sigma}_{\mathbf{u}})$, where $\mathbf{M}_{\mathbf{u}} \in \mathbb{R}^{M_H M_X}$, $\boldsymbol{\Sigma}_{\mathbf{u}} \in \mathbb{R}^{M_H M_X \times M_H M_X}$. Practically, instead of directly parametrizing $q(\mathbf{u})$, we introduce $\mathbf{u}_0$ with $p(\mathbf{u}_0) = \mathcal{N}(\mathbf{u}_0 \mid \mathbf{0}, \mathbb{I}_{M_H M_X})$, and assume $\mathbf{u} = \mathbf{L}\mathbf{u}_0$, where $\mathbf{L}\mathbf{L}^\top = \mathbf{K}_{\mathbf{u},\mathbf{u}}$. We parametrize $q(\mathbf{u}_0)$ as $\mathcal{N}(\mathbf{u}_0 \mid \mathbf{M}_0, \boldsymbol{\Sigma}_0^H \otimes \boldsymbol{\Sigma}_0^X)$, where $\boldsymbol{\Sigma}_0^H \in \mathbb{R}^{M_H \times M_H}$ and $\boldsymbol{\Sigma}_0^X \in \mathbb{R}^{M_X \times M_X}$. This procedure does not alter the prior distribution of $\mathbf{u}$ but reduces the parameters from $\mathcal{O}(M_H^2 M_X^2)$ to $\mathcal{O}(M_H^2 + M_X^2 + M_H M_X)$. [2] The variational posterior distribution for $\mathbf{f}$ becomes:

$$q(\mathbf{f} \mid \mathbf{H}, \mathbf{X}, \mathbf{Z}^{\mathbf{H}}, \mathbf{Z}^{\mathbf{X}}) = \int p(\mathbf{f} \mid \mathbf{u}, \mathbf{H}, \mathbf{X}, \mathbf{Z}^{\mathbf{H}}, \mathbf{Z}^{\mathbf{X}}) q(\mathbf{u}) d\mathbf{u}$$

$$= \mathcal{N}(\mathbf{f} \mid \mathbf{K}_{\mathbf{f},\mathbf{u}}\mathbf{K}_{\mathbf{u},\mathbf{u}}^{-1}\mathbf{M}_{\mathbf{u}}, \mathbf{K}_{\mathbf{f},\mathbf{f}} + \mathbf{K}_{\mathbf{f},\mathbf{u}}\mathbf{K}_{\mathbf{u},\mathbf{u}}^{-1}\boldsymbol{\Sigma}_{\mathbf{u}}\mathbf{K}_{\mathbf{u},\mathbf{u}}^{-1}\mathbf{K}_{\mathbf{u},\mathbf{f}} - \mathbf{K}_{\mathbf{f},\mathbf{u}}\mathbf{K}_{\mathbf{u},\mathbf{u}}^{-1}\mathbf{K}_{\mathbf{u},\mathbf{f}}).$$

## 3.3 Variational Lower Bound with Stochastic Optimisation

As shown previously in Eq. 5, the ELBO is defined as

$$\mathcal{L}_{\text{elbo}} = \underbrace{\mathbb{E}_{q(\mathbf{f}|\mathbf{X},\mathbf{H})q(\mathbf{H})}\left[\log p(\mathbf{Y} \mid \mathbf{f})\right]}_{\mathcal{F}} - \mathrm{KL}\left(q(\mathbf{u}) \,\|\, p(\mathbf{u})\right) - \mathrm{KL}\left(q(\mathbf{H}) \,\|\, p(\mathbf{H})\right), \tag{9}$$

where the $\mathcal{F}$ term is analytically integrated in LV-MOGP. However, this tractability is only feasible for Gaussian likelihoods. For non-Gaussian likelihoods, such as the Poisson likelihood, the $\mathcal{F}$ term must be re-derived and approximated. To facilitate support for non-Gaussian likelihoods and mini-batch gradient updates, we consider deriving the ELBO differently. Considering the factorisation: $\log p(\mathbf{Y} \mid \mathbf{f}) = \sum_{d=1}^D \sum_{n=1}^N \log p(y_{dn} \mid f_{dn})$, we obtain:

$$\mathcal{F} = \mathbb{E}_{q(\mathbf{f}|\mathbf{X},\mathbf{H})\,q(\mathbf{H})}\left[\sum_{d=1}^D \sum_{n=1}^N \log p(y_{dn} \mid f_{dn})\right] = \sum_{d=1}^D \sum_{n=1}^N \mathbb{E}_{q(\mathbf{H}_d)}\left[\underbrace{\mathbb{E}_{q(f_{dn}|\mathbf{H}_d, \mathbf{x}_n)}\left[\log p(y_{dn} \mid f_{dn})\right]}_{\mathcal{L}_{dn}(\mathbf{H}_d)}\right]$$

$$= \sum_{d=1}^D \sum_{n=1}^N \mathbb{E}_{q(\mathbf{H}_d)}\left[\mathcal{L}_{dn}(\mathbf{H}_d)\right], \tag{10}$$

and the expectation term $\mathbb{E}_{q(\mathbf{H}_d)}[\mathcal{L}_{dn}(\mathbf{H}_d)]$ will be computed numerically using Monte Carlo estimation with $J$ samples $\{\mathbf{H}_d^{(j)}\}_{j=1}^J = \{\mathbf{h}_{d,1}^{(j)}, \mathbf{h}_{d,2}^{(j)}, ..., \mathbf{h}_{d,Q}^{(j)}\}_{j=1}^J$, sampled from $q(\mathbf{h}_{d,1}), q(\mathbf{h}_{d,2}), ..., q(\mathbf{h}_{d,Q})$ using reparametrisation trick (Kingma & Welling, 2013; Lalchand et al., 2022). In particular, we sample $\epsilon^{(j)} \sim \mathcal{N}(\epsilon^{(j)} \mid \mathbf{0}, \mathbb{I}_{Q_H})$ and compute $\mathbf{h}_{d,q}^{(j)} = \mathbf{m}_{d,q} + \mathbf{S}_{d,q} \odot \epsilon^{(j)}$ for $q \in \{1, 2, ..., Q\}$ and $j \in \{1, 2, ..., J\}$. Thus,

$$\mathbb{E}_{q(\mathbf{H}_d)}[\mathcal{L}_{dn}(\mathbf{H}_d)] \approx \frac{1}{J}\sum_{j=1}^J \mathcal{L}_{dn}(\mathbf{H}_d^{(j)}) = \frac{1}{J}\sum_{j=1}^J \mathcal{L}_{dn}\left(\left\{\mathbf{m}_{d,q} + \mathbf{S}_{d,q} \odot \epsilon^{(j)}\right\}_{q=1}^Q\right), \tag{11}$$

---

[2]Other benefits such as efficient computation of KL term are detailed in Appendix A.2.

where $\odot$ denotes the Hadamard product. For a Gaussian likelihood, the expected log-likelihood term $\mathcal{L}_{dn}(\mathbf{H}_d^{(j)})$ can be analytically obtained,

$$
\mathcal{L}_{dn}(\mathbf{H}_d^{(j)}) = \log \mathcal{N}(y_{dn} \mid \mathbf{K}_{f_{dn},\mathbf{u}}\mathbf{K}_{\mathbf{u},\mathbf{u}}^{-1}\mathbf{M}_\mathbf{u}, \sigma_d^2) - \frac{1}{2\sigma_d^2}\mathrm{Tr}(\mathbf{K}_{f_{dn},f_{dn}}) + \frac{1}{2\sigma_d^2}\mathrm{Tr}(\mathbf{K}_{\mathbf{u},\mathbf{u}}^{-1}\mathbf{K}_{\mathbf{u},f_{dn}}\mathbf{K}_{f_{dn},\mathbf{u}})
$$
$$
- \frac{1}{2\sigma_d^2}\mathrm{Tr}(\mathbf{\Sigma}_\mathbf{u}\mathbf{K}_{\mathbf{u},\mathbf{u}}^{-1}\mathbf{K}_{\mathbf{u},f_{dn}}\mathbf{K}_{f_{dn},\mathbf{u}}\mathbf{K}_{\mathbf{u},\mathbf{u}}^{-1}). \tag{12}
$$

For non-Gaussian likelihood, this one-dimensional integral can be accurately approximated by Gauss-Hermite quadrature (Liu & Pierce, 1994; Ramchandran et al., 2021), see Appendix A.3.

**Doubly Stochastic ELBO**   We further approximate $\mathcal{L}_{elbo}$ by employing mini-batching to speed up computation. In every iteration, a minibatch of $m_b$ input-output pairs is sampled, denoted as $\mathcal{B} = \{(d_1, n_1), (d_2, n_2), ..., (d_{m_b}, n_{m_b})\}$,

$$
\hat{\mathcal{L}}_{elbo} = \frac{ND}{m_b} \sum_{(d,n)\in\mathcal{B}} \frac{1}{J} \sum_{j=1}^{J} \mathbb{E}_{q(f_{dn}\mid\mathbf{H}_d^{(j)},\mathbf{x}_n)} \left[\log p(y_{dn} \mid f_{dn})\right] - \mathrm{KL}(q(\mathbf{u})||p(\mathbf{u})) - \frac{D}{m_b} \sum_{i=1}^{m_b} \sum_{q=1}^{Q} \mathrm{KL}(q(\mathbf{h}_{d_i,q})||p(\mathbf{h}_{d_i,q})),
$$

The KL terms are analytically tractable due to the choice of the Gaussian variational family for $q(\mathbf{u})$ and $q(\mathbf{h}_{d_i,q})$. This method is referred to as *doubly stochastic variational inference* (Titsias & Lázaro-Gredilla, 2014; Salimbeni & Deisenroth, 2017), reflecting the two-fold stochasticity: mini-batching for gradient-based updates and computing expectation via Monte Carlo sampling. This training procedure factorises the data-dependent term across data points, allowing trivial marginalisation of *missing values*. Consequently, the ELBO is naturally compatible with the *heterotopic*[3] setting.

**Computational Complexity**   The training cost for GS-LVMOGP is primarily dominated by two operations: matrix inversion $\mathbf{K}_{\mathbf{u},\mathbf{u}}^{-1}$ and matrix multiplication $\mathbf{K}_{\mathbf{f}_b,\mathbf{u}}\mathbf{K}_{\mathbf{u},\mathbf{u}}^{-1}$, where $\mathbf{f}_b$ represents the $m_b$ function values in a mini-batch. The matrix multiplication has computational complexity $\mathcal{O}(m_b M_X^2 M_H^2)$. The computational complexity for matrix inversion $\mathbf{K}_{\mathbf{u},\mathbf{u}}^{-1}$ varies based on the choice of $Q$. For $Q = 1$, the matrix $\mathbf{K}_{\mathbf{u},\mathbf{u}}$ has the a Kronecker product structure, allowing the inversion $\mathbf{K}_{\mathbf{u},\mathbf{u}}^{-1} = (\mathbf{K}_{\mathbf{u},\mathbf{u}}^H)^{-1} \otimes (\mathbf{K}_{\mathbf{u},\mathbf{u}}^X)^{-1}$ to be performed in $\mathcal{O}(M_X^3 + M_H^3)$. For $Q > 1$, inversion operation $\mathbf{K}_{\mathbf{u},\mathbf{u}}^{-1}$ has a computational complexity $\mathcal{O}(M_X^3 M_H^3)$. Compared to LV-MOGP, the GS-LVMOGP has a computational complexity that is free from dependence on the size of the outputs $D$ and inputs $N$. This makes the method more capable of handling large-scale problems. [4] However, it is worth noting that the scalability of GS-LVMOGP is limited by the number of inducing variables $M_H M_X$. Additionally, unlike LV-MOGP, which uses a single set of inducing points in the latent space, GS-LVMOGP employs $Q$ sets of inducing points. This difference increases the computational burden as $Q$ grows. Despite this, the flexibility of using multiple inducing points across different latent spaces allows GS-LVMOGP to capture more complex output structures, making it still suitable for large-scale problems, though this added complexity should be considered as a trade-off.

### 3.4   Prediction

After model training, we can make predictions for a new input $\mathbf{x}^*$ at any output $d^*$. The predictive distribution for $f^*$ is given by:

$$
p(f^* \mid \mathbf{Z}^H, \mathbf{Z}^X, \mathbf{x}^*) = \int p(f^* \mid \mathbf{Z}^H, \mathbf{Z}^X, \mathbf{H}_{d^*}, \mathbf{x}^*) q(\mathbf{H}_{d^*}) d\mathbf{H}_{d^*}
$$

---

[3]For MOGP, if each output has the same set of inputs, the system is known as *isotopic*. For general cases, the outputs may be associated with different sets of inputs, $\mathbf{X}_d = \{\mathbf{x}_{dn}\}_{n=1}^{N_d}$ this is known as *heterotopic*.

[4]We are focused on computational complexity for each iteration of the parameter update. Though the smaller mini-batch size $m_b$ will lead to smaller computational complexity per iteration, more iterations are required for cycling through all the data. Furthermore, a smaller learning rate is often required for small $m_b$ to tradeoff the larger noise in the ELBO approximation. Therefore, there is a complex relationship between learning rate and minibatch size which determines the true computational complexity.

However, Eq. 13 is intractable for general kernel functions $k_q^H$ [5]. As a first workaround, we can approximate $q(\mathbf{H}_{d^*}) = \prod_{q=1}^{Q} q(\mathbf{h}_{d^*q})$ by its $Q$ means, where $\mathbf{h}_{d^*,q}^{mean}$ denotes the mean of $q(\mathbf{h}_{d^*,q})$. Thus, $p(f^* \mid \mathbf{Z}^H, \mathbf{Z}^X, \mathbf{x}^*) \approx p(f^* \mid \{\mathbf{h}_{d^*,q}^{mean}\}_{q=1}^Q, \mathbf{Z}^H, \mathbf{Z}^X, \mathbf{x}^*)$. We use this approach for our experiments.

## 4 Related Works

There have been many suggestions regarding the construction of MOGP models. One line of work (Boyle & Frean, 2004; Alvarez & Lawrence, 2008; 2011) convolves smoothing kernels with a set of latent GP functions to produce correlated outputs. Despite the elegance, they have to choose friendly (but perhaps less expressive) kernels, such as Gaussian and delta, to ensure analytical convolution results (Zhe et al., 2019). To enhance the capability for multitask modelling, one can use spectral mixture kernels (Wilson & Adams, 2013; Parra & Tobar, 2017; Altamirano & Tobar, 2022), which are able to model phase differences and delays among channels for more expressive modelling. Collaborative MOGP (Nguyen et al., 2014) assumes that the target values for each output are composed of two components: latent GPs that capture the shared structure across outputs, and individual GPs that model the residual or output-specific variations. Another type of method (Higdon et al., 2008; Xing et al., 2015; 2016) assume outputs can be formed by a linear combination of fixed bases, which can efficiently handle a large number of outputs. Bruinsma et al. (2020) proposes an approach to use orthogonal bases to decouple the latent functions, further accelerating the inference and learning process. For the isotopic data setting, the Kronecker product structured kernel matrices can be used for MOGPs (Rakitsch et al., 2013; Bonilla et al., 2007; Stegle et al., 2011), whose algebraic properties are exploited for fast training and inference. The MOGP model can be reformulated sequentially into a set of conditioned univariate GPs with previous outputs incorporated as additional inputs (García-Hinde et al., 2022), thereby simplifying the training process. This concept is closely connected to autoregressive modelling for MOGPs Requeima et al. (2019).

There are many recent advances regarding LMC in the MOGP literature. Moreno-Muñoz et al. (2018) extends classic LMC (Journel & Huijbregts, 1976; Goovaerts, 1997) to address heterogeneous regression tasks, enabling each output to be associated with a (possibly) distinct likelihood function. Giraldo & Alvarez (2021) proposes a fully natural gradient scheme to improve the heterogeneous MOGP for both LMC and process convolution. Liu et al. (2022) proposes the use of neural embeddings to project latent independent GPs into a higher-dimensional and more diverse space, thereby increasing the modelling capacity for LMC. Yoon et al. (2022) proposes a special case of LMC by fixing some of the kernel or coregionalization matrices to identity or all-one matrix, with the goal of decomposing input effects on outputs into components shared across or specific to tasks and samples. By leveraging the equivalence between certain temporal GPs and Stochastic Differential Equations (SDEs) (Särkkä & Solin, 2019), Jeong & Kim (2023) represent the latent GPs in the LMC as factorial SDEs. This reformulation enables the application of a range of techniques (Adam et al., 2020), which reduce the training time complexity to scale linearly with respect to the number of samples or inducing points.

## 5 Experiments

We test the GS-LVMOGP on several real-world data sets. [6] For all experiments, we choose automatic relevance determination squared exponential (SE-ARD) kernel on the latent space. The kernel of GS-LVMOGP on the input space is different for each dataset and is specified accordingly. More information about evaluation metrics and experiment details, including the values for $M_H$, $M_X$, $m_b$, and $Q_H$ are in Appendix A.6. The implementation of our model can be found in `https://github.com/XiaoyuJiang17/GS-LVMOGP`.

**Exchange Rates prediction**   This dataset includes daily exchange rates against the USD for ten currencies and three precious metals for the year 2007. Our task is to predict the exchange rates of CAD, JPY, and AUD on specific days, given that all other currencies are observed throughout the year. We follow the same setup as Álvarez et al. (2010). For GS-LVMOGP models, we use a Matérn-1/2 kernel, consistent with

---

[5]Further discussion on potentially how to handle this integral appears in Appendix A.5.
[6]And a synthetic dataset in Appendix A.6.1.

Table 1: Methods comparison on Exchange dataset. IGP denotes independent GP, one for each output. COGP (Nguyen et al., 2014), CGP (Alvarez & Lawrence, 2008), OILMM (Bruinsma et al., 2020), HetMOGP (Moreno-Muñoz et al. (2018)). * Numbers are taken from Nguyen et al. (2014), † Numbers are taken from Bruinsma et al. (2020). Results are averages of the outputs over five repetitions with different random seeds.

| Model | IGP | COGP | CGP | OILMM | HetMOGP | LV-MOGP | GS-LVMOGP |  |  |
|---|---|---|---|---|---|---|---|---|---|
| | | | | | | | $Q=1$ | $Q=2$ | $Q=3$ |
| SMSE | 0.600* | 0.213* | 0.243* | 0.19† | 0.92 | 0.251 | 0.256 | 0.186 | **0.167** |
| NLPD | 0.408* | $-0.839^*$ | $-\mathbf{2.947^*}$ | | -1.3 | -2.471 | -1.851 | -2.416 | -2.703 |

Bruinsma et al. (2020). Though in this experiment the number of outputs is rather small, $D = 13$, and no approximation is required, we included it as a way to show that the mini-batch approach leads to results on par with other small-scale models, as shown in Table 1. Notice that by setting $Q = 1$, we obtain a mini-batch version of the LV-MOGP proposed by Dai et al. (2017). Though its performance is slightly behind LV-MOGP, the GS-LVMOGP with $Q = 3$ outperforms LV-MOGP on both metrics. As $Q$ increases, the flexibility in constructing the covariance matrix improves, resulting in enhanced model performance. More information is elaborated in Appendix A.6.2.

**NYC Crime Count modelling**  We analyse crime patterns in New York City (NYC) using daily complaint data from NYC Crimes2014. Accurate modelling of the seasonal trends and spatial dependencies from crime data can improve police resource allocation efficiency (Aglietti et al., 2019). Following Hamelijnck et al. (2021), the dataset includes 447 spatial locations with 182 observations each. Each location is treated as an output, so $D = 447$. We consider three models: IGP, OILMM and our GS-LVMOGP, all using the Matérn-3/2 kernel. We consider Gaussian and Poisson likelihoods for this count data. The construction of OILMM and exact GP heavily relies on the Gaussian likelihood. We instead consider independent SVGPs (Titsias, 2009; Hensman et al., 2013) equipped with the Poisson likelihood and 1000 inducing points for each output as a baseline. Table 2 shows the results. The results for GS-LVMOGP with $Q = 1, 2, 3$ again indicate that higher $Q$ generally improves performance. Notably, the Poisson likelihood consistently outperforms the Gaussian likelihood, as it is inherently more suited to the characteristics of count data.

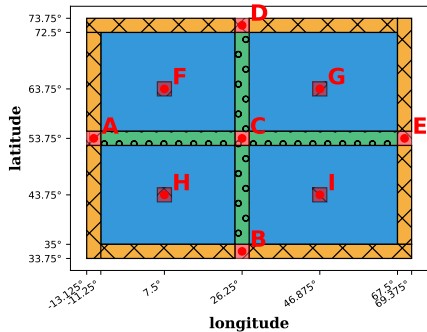

Figure 1: Spatial locations for training (blue), inner (green) and outer (orange) output extrapolation. Predictions for I, A and C are in Fig. 2, and others (B, D, E, F, G, H) are in Appendix A.6.3

Table 2: Models comparison on NYC Crime. The brackets (G) and (P) refer to Gaussian and Poisson likelihood, respectively. Results are averages over 5-fold cross-validations with ± standard deviation.

| | | RMSE | NLPD |
|---|---|---|---|
| IGP (G) | | 1.937±0.030 | 2.107±0.022 |
| I-SVGP-1000 (P) | | 3.000±0.035 | 1.939±0.014 |
| OILMM (G) | | 1.857±0.025 | 1.493±0.011 |
| | $Q=1$ | 2.201±0.125 | 1.498±0.299 |
| GS-LVMOGP (G) | $Q=2$ | 1.908±0.381 | 1.525±0.206 |
| | $Q=3$ | 1.969±0.103 | 1.531±0.352 |
| | $Q=1$ | 1.791±0.023 | 1.288±0.003 |
| GS-LVMOGP (P) | $Q=2$ | 1.791±0.023 | **1.287 ± 0.003** |
| | $Q=3$ | **1.790 ± 0.024** | **1.287 ± 0.003** |

**Spatiotemporal Temperature modelling**  In this section, we address spatiotemporal temperature modelling across Europe, using $1,260$ spatial locations (blue regions in Fig. 1) with 363 months of observations per location. Each spatial location is treated as an output, so $D = 1260$, and time $t$ is the input. Our tasks include data imputation and extrapolation prediction. For each output, we randomly select 10 data points from the first 263 observations as training data, using the remaining 253 months for imputation testing. The last 100 observations for each output serve as extrapolation test samples. To incorporate spatial information, we set the means of the prior distribution of the latent variables, $p(\mathbf{h}_{d,q}), q = 1, 2, ..., Q$, to the (longitude,

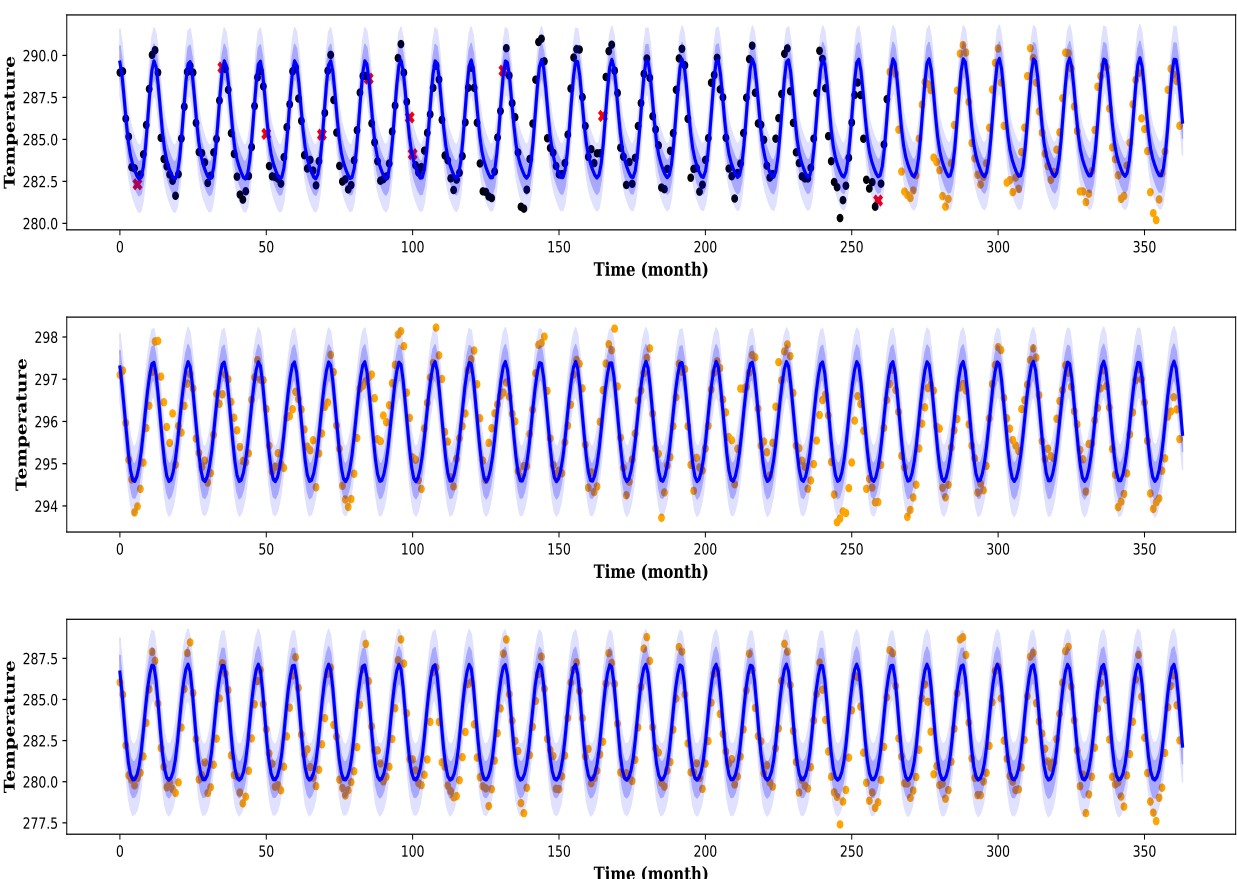

Figure 2: Model predictions are provided for three selected outputs corresponding to specific spatial locations. The temperature is measured in Kelvin units. From top to bottom, the plots represent predictions at locations I, A and C as marked in Fig. 1. Location I is included in the training dataset, while predictions for locations A and C are the result of output extrapolation. Training points are indicated by red crosses (✗), imputation and extrapolation test points by black (●) and orange dots (●) respectively. The shaded area indicates the mean ± one and two standard deviations.

latitude) vector for each output $d$. We compare three models: IGP, OILMM and GS-LVMOGP, all using Matérn–5/2 kernels with a periodic component. Table 3 summarises the results. From Table 3, with $Q = 3$, the GS-LVMOGP outperforms other methods in both imputation and extrapolation tasks. The first plot in Fig. 2 shows predictions for one output in this spatiotemporal dataset. Our model can also extrapolate to unseen locations not included during training, known as *output extrapolation*, by setting latent variables $\mathbf{h}_{d^*,q}$ to the spatial coordinates of chosen locations. The last two plots in Fig. 2 show predictions for two spatial locations excluded during training. Fig. 1 illustrates two types of spatial regions for output extrapolation, marked in green and orange. Locations marked in green are surrounded by training locations (blue), thus the predictions for them are termed as *inner output extrapolation*, a total of 69 outputs. In contrast, predictions for the orange regions, referred to as *outer output extrapolation*, lie on the periphery of the training regions, totalling 144 outputs. The SMSEs for *inner* and *outer* output extrapolation are $0.184 \pm 0.09$ and $0.211 \pm 0.23$, respectively. [7]

**Climate forecast**    We consider the United States Historical Climatology Network (USHCN) daily data set, and we follow a similar setting to De Brouwer et al. (2019), choosing a subset of $1,114$ stations and examining a four-year observational period from 1996 to 2000. The dataset is subsampled as De Brouwer et al. (2019), resulting in on average 52 observations during the first three years for each output. We discard outputs with

---

[7]The NLPDs for extrapolated outputs are not available as we have no estimates of the parameters $\sigma_d$.

Table 3: Comparison of methods on the spatio-temporal dataset. The results are averages over five repetitions with different random seeds, with the standard deviation in the bracket.

| | | IGP | OILMM | GS-LVMOGP | | |
| --- | --- | --- | --- | --- | --- | --- |
| | | | | $Q = 1$ | $Q = 2$ | $Q = 3$ |
| Imputation | SMSE | 0.177 (1.1e-3) | 0.128 (0.13) | 0.135 (3.9e-3) | 0.123 (3e-3) | **0.120** (1.8e-3) |
| | NLPD | 2.484 (1.1e-2) | 2.85 (0.26) | 2.42 (0.02) | **2.380** (1.2e-2) | **2.380** (6.6e-3) |
| Extrapolation | SMSE | 0.261 (7e-3) | 0.147 (0.12) | 0.146 (5.6e-3) | 0.137 (2.6e-3) | **0.133** (3.5e-3) |
| | NLPD | 2.87 (0.03) | 3.11 (0.22) | 2.568 (0.03) | 2.565 (8.1e-3) | **2.558** (2.5e-2) |

Table 4: Model comparisons on USHCN. Results are averages over 5 repetitions with different random seeds, with mean $\pm$ standard deviation.

| | | MSE | NLPD |
| --- | --- | --- | --- |
| GS-LVMOGP | $Q = 1$ | $0.620 \pm 0.12$ | $\mathbf{8.747} \pm 1.8$ |
| | $Q = 2$ | $0.619 \pm 0.09$ | $10.16 \pm 1.7$ |
| | $Q = 3$ | $\mathbf{0.618} \pm 0.08$ | $9.89 \pm 1.8$ |
| OILMM | | $0.89 \pm 0.02$ | $810.37 \pm 1.5$ |

fewer than three observations and finally have $5,507$ outputs. More details are in Appendix A.6.5. Our task is to forecast the subsequent three observations following the initial three years of data collection. We use the Matérn-3/2 kernel for GS-LVMOGP and OILMM. Table 4 compares the performance results of our models against OILMM, showing improved performance.

**Spatial Transcriptomics** Spatial transcriptomics (Ståhl et al., 2016) offers high-resolution profiling of gene expression while retaining the spatial information of the tissue. Applying machine learning techniques to spatial transcriptomics datasets is vital for understanding the tissue and the disease architecture, potentially enhancing diagnosis and treatment. We consider a 10x Genomics Visium human prostate cancer dataset

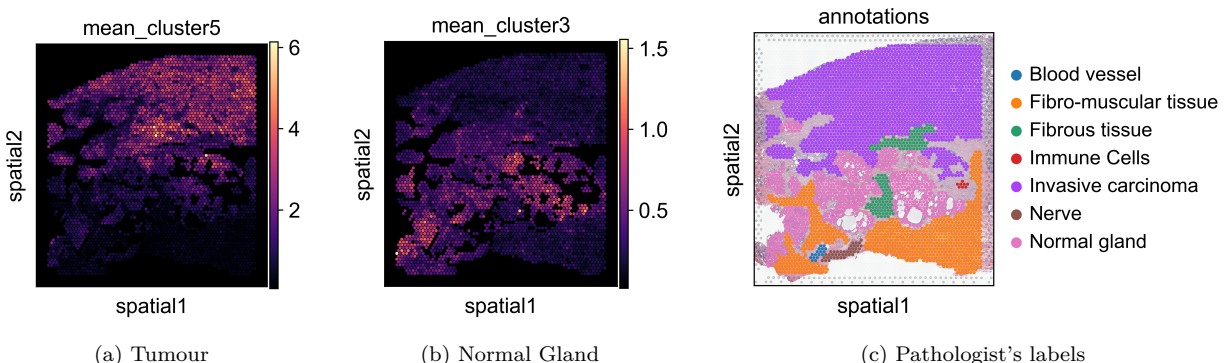

(a) Tumour          (b) Normal Gland         (c) Pathologist's labels

Figure 3: Latent variable *k-means* clustering results of the prostate carcinoma dataset. (a) and (b) show the average gene expression in clusters 5 and 3, respectively. (c) shows the ground-truth pathologist's annotations. Cluster 5 aligns well with the invasive carcinoma label (tumour) while cluster 3 aligns with the normal gland (normal). Plots for other clusters are shown in Appendix A.6.6.

(10XGenomics2024) which contains gene expression counts data from $17,943$ genes measured across $4,371$ spatial locations. Pathologist's histological annotations for this specific tissue. Fig. 3c are provided by 10x Genomics. For our model, we use the top $5,000$ highly variable genes calculated using Scanpy's highly variable genes function (Wolf et al., 2018). Each gene is treated as a different output, and the spatial coordinates of cells are regarded as the inputs. We focus on GS-LVMOGP with $Q = 1$, using the SE-ARD kernel and Poisson likelihood, and we aim to explore the structure of the latent space with $Q_H = 3$ for the genes.

After fitting our model to this dataset, we collect the latent variables for the genes and cluster them into 6 groups using *k-means*. To plot the spatial distribution of each cluster onto the tissue, we calculate the average expression of the genes in each of the clusters and compare them against the pathologist's annotated regions. In Fig. 3 we show two of the clusters with genes delineating the tumour (3b) and the normal tissue (3c) areas, showing good correspondence with the pathologist's labels (Invasive carcinoma and Normal glands).

Unlike standard clustering practices which solely rely on gene expression to cluster the spatial transcriptomics data, our clustering approach incorporates spatial correlations into the gene clusters. In this way, we can discover distinct spatial tissue regions which might be missed when only considering gene expression.

## 6   Conclusions

In this paper, we propose GS-LVMOGP, a generalised latent variable multi-output Gaussian process model within a stochastic variational inference framework. Our approach extends the Latent Variable Multi-Output Gaussian Process (LV-MOGP) model (Dai et al., 2017), which is analogous to the Intrinsic Coregionalisation Model (ICM) due to its use of a single coregionalisation matrix ($Q = 1$). By generalising this framework to allow multiple coregionalisation matrices ($Q > 1$), we introduce additional flexibility into the covariance structure, which enables different kernels to act on different latent spaces $H$ to capture various types of correlation structure among outputs, potentially through the use of varying lengthscales. By performing variational inference for latent variables $q(\mathbf{H})$ and inducing values $q(\mathbf{u})$, our approach can effectively manage large-scale datasets with Gaussian and non-Gaussian likelihoods. One feature of the model is that the parameters in the mean vectors and variances for $q(\mathbf{H})$ also increase with the number of outputs. Future research could explore imposing structured constraints in the latent space to further reduce the number of parameters required for estimation.

## Acknowledgments

We thank the TMLR editor and reviewers for constructive feedback, and sincerely appreciate Xinxing Shi for insightful and pleasant discussions throughout the progress of this work. Mauricio A. Álvarez has been financed by the EPSRC Research Projects EP/R034303/1, EP/T00343X/2, EP/V029045/1, the UKRI cross-council grant MR/Z505468/1, and the Wellcome Trust project 217068/Z/19/Z. Xiaoyu Jiang is supported by the University of Manchester Departmental Studentship for the Department of Computer Science.

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

# A  Appendix

## A.1  ELBO deviation for LV-MOGP

In this section, we describe in more detail the deviation of the evidence lower bound for LV-MOGP.

$$
\begin{aligned}
\log p(\mathbf{Y} \mid \mathbf{X}) &= \log \int p(\mathbf{Y}, \mathbf{f}, \mathbf{u}, \mathbf{H} \mid \mathbf{X}) d\mathbf{f} d\mathbf{u} d\mathbf{H} \\
&= \log \int p(\mathbf{Y}, \mathbf{f}, \mathbf{u}, \mathbf{H} \mid \mathbf{X}) \frac{q(\mathbf{f}, \mathbf{u}, \mathbf{H})}{q(\mathbf{f}, \mathbf{u}, \mathbf{H})} d\mathbf{f} d\mathbf{u} d\mathbf{H} \\
&\geq \int q(\mathbf{f}, \mathbf{u}, \mathbf{H}) \log \frac{p(\mathbf{Y}, \mathbf{f}, \mathbf{u}, \mathbf{H} \mid \mathbf{X})}{q(\mathbf{f}, \mathbf{u}, \mathbf{H})} d\mathbf{f} d\mathbf{u} d\mathbf{H} \\
&= \int q(\mathbf{f}, \mathbf{u}, \mathbf{H}) \log \frac{p(\mathbf{Y} \mid \mathbf{f}) \cancel{p(\mathbf{f} \mid \mathbf{u})} p(\mathbf{u}) p(\mathbf{H})}{\cancel{p(\mathbf{f} \mid \mathbf{u})} q(\mathbf{u}) q(\mathbf{H})} d\mathbf{f} d\mathbf{H} \\
&= \underbrace{\int q(\mathbf{f}, \mathbf{u}, \mathbf{H}) \log p(\mathbf{Y} \mid \mathbf{f}) d\mathbf{f} d\mathbf{u} d\mathbf{H}}_{\mathcal{F}} + \int q(\mathbf{u}) \log \frac{p(\mathbf{u})}{q(\mathbf{u})} d\mathbf{u} + \int q(\mathbf{H}) \log \frac{p(\mathbf{H})}{q(\mathbf{H})} d\mathbf{H} \\
&= \mathcal{F} - \mathrm{KL}(q(\mathbf{u}) || p(\mathbf{u})) - \mathrm{KL}(q(\mathbf{H}) || p(\mathbf{H})).
\end{aligned}
\tag{13}
$$

In the third row, we use Jensen's inequality. Now we focus on the $\mathcal{F}$ term. Notice that:

$$
\log p(\mathbf{Y} \mid \mathbf{f}) = \log \mathcal{N}(\mathbf{Y} \mid \mathbf{f}, \sigma^2 \mathbb{I}) = -\frac{ND}{2} \log(2\pi\sigma^2) - \frac{1}{2\sigma^2} \left( \mathbf{Y}^\top \mathbf{Y} - 2\mathbf{Y}^\top \mathbf{f} + \mathbf{f}^\top \mathbf{f} \right),
\tag{14}
$$

$$
q(\mathbf{f}) = \int q(\mathbf{f} \mid \mathbf{u}) q(\mathbf{u}) d\mathbf{u} = \mathcal{N}(\mathbf{f} \mid \mathbf{K}_{\mathbf{f},\mathbf{u}} \mathbf{K}_{\mathbf{u},\mathbf{u}}^{-1} \mathbf{M}^{\mathbf{u}}, \mathbf{K}_{\mathbf{f},\mathbf{f}} + \mathbf{K}_{\mathbf{f},\mathbf{u}} \mathbf{K}_{\mathbf{u},\mathbf{u}}^{-1} \mathbf{\Sigma}^{\mathbf{u}} \mathbf{K}_{\mathbf{u},\mathbf{u}}^{-1} \mathbf{K}_{\mathbf{u},\mathbf{f}} - \mathbf{K}_{\mathbf{f},\mathbf{u}} \mathbf{K}_{\mathbf{u},\mathbf{u}}^{-1} \mathbf{K}_{\mathbf{u},\mathbf{f}}),
\tag{15}
$$

$$
\begin{aligned}
\mathcal{F} &= \int q(\mathbf{H}) \int q(\mathbf{f}) \log p(\mathbf{Y} \mid \mathbf{f}) d\mathbf{f} d\mathbf{H} \\
&= \int q(\mathbf{H}) \left[ -\frac{ND}{2} \log 2\pi\sigma^2 - \frac{1}{2\sigma^2} \mathbf{Y}^\top \mathbf{Y} + \frac{1}{\sigma^2} \mathbf{Y}^\top \mathbb{E}_{q(\mathbf{f})}[\mathbf{f}] - \frac{1}{2\sigma^2} \mathbb{E}_{q(\mathbf{f})}[\mathbf{f}^\top \mathbf{f}] \right] d\mathbf{H} \\
&= -\frac{ND}{2} \log 2\pi\sigma^2 - \frac{1}{2\sigma^2} \mathbf{Y}^\top \mathbf{Y} + \int q(\mathbf{H}) \left[ \frac{1}{\sigma^2} \mathbf{Y}^\top \mathbf{K}_{\mathbf{f},\mathbf{u}} \mathbf{K}_{\mathbf{u},\mathbf{u}}^{-1} \mathbf{M}^{\mathbf{u}} \right] d\mathbf{H} - \int q(\mathbf{H}) \frac{1}{2\sigma^2} \\
&\qquad \left[ \mathrm{Tr}(\mathbf{K}_{\mathbf{f},\mathbf{f}} - \mathbf{K}_{\mathbf{u},\mathbf{u}}^{-1} \mathbf{K}_{\mathbf{u},\mathbf{f}} \mathbf{K}_{\mathbf{f},\mathbf{u}}) + \mathrm{Tr}(\mathbf{K}_{\mathbf{u},\mathbf{u}}^{-1} \mathbf{K}_{\mathbf{u},\mathbf{f}} \mathbf{K}_{\mathbf{f},\mathbf{u}} \mathbf{K}_{\mathbf{u},\mathbf{u}}^{-1} (\mathbf{M}^{\mathbf{u}} (\mathbf{M}^{\mathbf{u}})^\top + \mathbf{\Sigma}^{\mathbf{u}})) \right] d\mathbf{H} \\
&= -\frac{ND}{2} \log 2\pi\sigma^2 - \frac{1}{2\sigma^2} \mathbf{Y}^\top \mathbf{Y} - \frac{1}{2\sigma^2} \mathrm{Tr}(\mathbf{K}_{\mathbf{u},\mathbf{u}}^{-1} \Phi \mathbf{K}_{\mathbf{u},\mathbf{u}}^{-1} (\mathbf{M}^{\mathbf{u}} (\mathbf{M}^{\mathbf{u}})^\top + \mathbf{\Sigma}^{\mathbf{u}})) + \frac{1}{\sigma^2} \mathbf{Y}^\top \Psi \mathbf{K}_{\mathbf{u},\mathbf{u}}^{-1} \mathbf{M}^{\mathbf{u}} \\
&\qquad - \frac{1}{2\sigma^2} (\psi - \mathrm{Tr}(\mathbf{K}_{\mathbf{u},\mathbf{u}}^{-1} \Phi)),
\end{aligned}
\tag{16}
$$

where $\psi = \langle \mathrm{Tr}(\mathbf{K}_{\mathbf{f},\mathbf{f}}) \rangle_{q(\mathbf{H})}$, $\Psi = \langle \mathbf{K}_{\mathbf{f},\mathbf{u}} \rangle_{q(\mathbf{H})}$ and $\Phi = \langle \mathbf{K}_{\mathbf{u},\mathbf{f}} \mathbf{K}_{\mathbf{f},\mathbf{u}} \rangle_{q(\mathbf{H})}$. Notice that if there are no missing values, Eq. 16 can be further simplified by exploiting properties of the Kronecker product, resulting in equation (8) in Dai et al. (2017). But in the general case (with missing values), we use Eq. 16 to compute $\mathcal{F}_d$ for each output $d$, and $\mathcal{F} = \sum_{d=1}^{D} \mathcal{F}_d$.

### A.1.1 Computation of statistics $\psi$, $\Psi$ and $\Phi$

The statistics $\psi, \Psi$, and $\Phi$ can be simplified by exploiting the Kronecker product structure.

$$
\psi = \left\langle \mathrm{Tr}(\mathbf{K}_{\mathbf{f},\mathbf{f}}^H \otimes \mathbf{K}_{\mathbf{f},\mathbf{f}}^X) \right\rangle_{q(\mathbf{H})} = \left\langle \mathrm{Tr}(\mathbf{K}_{\mathbf{f},\mathbf{f}}^H) \right\rangle_{q(\mathbf{H})} \otimes \mathrm{Tr}(\mathbf{K}_{\mathbf{f},\mathbf{f}}^X) = \left( \sum_{d=1}^{D} \underbrace{\left\langle k^H(\mathbf{h}_d, \mathbf{h}_d) \right\rangle_{q(\mathbf{h}_d)}}_{\psi_d^H} \right) \otimes \mathrm{Tr}(\mathbf{K}_{\mathbf{f},\mathbf{f}}^X).
$$

$$
\Psi = \left\langle \mathbf{K}_{\mathbf{f},\mathbf{u}} \right\rangle_{q(\mathbf{H})} = \left\langle \mathbf{K}_{\mathbf{f},\mathbf{u}}^H \otimes \mathbf{K}_{\mathbf{f},\mathbf{u}}^X \right\rangle_{q(\mathbf{H})} = \underbrace{\left\langle \mathbf{K}_{\mathbf{f},\mathbf{u}}^H \right\rangle_{q(\mathbf{H})}}_{\Psi^H} \otimes \mathbf{K}_{\mathbf{f},\mathbf{u}}^X.
$$

$$
\Phi = \left\langle \mathbf{K}_{\mathbf{u},\mathbf{f}} \mathbf{K}_{\mathbf{f},\mathbf{u}} \right\rangle_{q(\mathbf{H})} = \left\langle \left( \mathbf{K}_{\mathbf{u},\mathbf{f}}^H \otimes \mathbf{K}_{\mathbf{u},\mathbf{f}}^X \right) \left( \mathbf{K}_{\mathbf{f},\mathbf{u}}^H \otimes \mathbf{K}_{\mathbf{f},\mathbf{u}}^X \right) \right\rangle_{q(\mathbf{H})} = \underbrace{\left\langle \mathbf{K}_{\mathbf{u},\mathbf{f}}^H \mathbf{K}_{\mathbf{f},\mathbf{u}}^H \right\rangle_{q(\mathbf{H})}}_{\Phi^H} \otimes \left( \mathbf{K}_{\mathbf{u},\mathbf{f}}^X \mathbf{K}_{\mathbf{f},\mathbf{u}}^X \right).
$$

(17)

The statistics $\psi_d^H$, $\Psi^H$, and $\Phi^H$ can be approximated by Monte Carlo methods. For some particular kernels, they can be analytically solved. For instance, for the SE-ARD kernel. Recall $q(\mathbf{h}_d) = \mathcal{N}(\mathbf{h}_d \mid \mathbf{M}_d, \mathbf{\Sigma}_d)$, with $\mathbf{\Sigma}_d$ being diagonal matrix, $s_{d,i}, i \in \{1, 2, ..., Q_H\}$ denotes the elements on the diagonal. $i$ th component of $\mathbf{M}_d$ is denoted as $m_{d,i}$. We are willing to derive analytical formulae for $\psi_d^H$, $\Psi_d^H = \left\langle \mathbf{K}_{f_d,\mathbf{u}}^H \right\rangle_{q(\mathbf{h}_d)}$ and $\Phi_d^H = \left\langle \mathbf{K}_{\mathbf{u},f_d}^H \mathbf{K}_{f_d,\mathbf{u}}^H \right\rangle_{q(\mathbf{h}_d)}$, notice that $\left\langle \mathbf{K}_{\mathbf{u},f_d}^H \mathbf{K}_{f_{d'},\mathbf{u}}^H \right\rangle_{q(\mathbf{H})} = \left\langle \mathbf{K}_{\mathbf{u},f_d}^H \right\rangle_{q(\mathbf{h}_d)} \left\langle \mathbf{K}_{f_{d'},\mathbf{u}}^H \right\rangle_{q(\mathbf{h}_{d'})} = (\Psi_d^H)^\top \Psi_{d'}^H$ if $d \neq d'$.

Recall for SE-ARD kernel, for any $\mathbf{h}_1, \mathbf{h}_2 \in \mathbb{R}^{Q_H}$:

$$
k^H(\mathbf{h}_1, \mathbf{h}_2) = \sigma_H^2 \exp(-\frac{1}{2} \sum_{i=1}^{Q_H} \frac{(\mathbf{h}_{1,i} - \mathbf{h}_{2,i})^2}{l_i}) = \underbrace{(2\pi)^{\frac{Q_H}{2}} \sigma_H^2 \prod_{i=1}^{Q_H} l_i^{\frac{1}{2}}}_{c} \mathcal{N}(\mathbf{h}_1 \mid \mathbf{h}_2, \mathrm{Diag}(l)),
$$

(18)

where $l \in \mathbb{R}^{Q_H}$ and $l_i$ is the $i$th component of $l$. The term $\psi_d^H$ is trivially computed as $c$.

Consider latent inducing variables $\mathbf{z}_i^H, \mathbf{z}_j^H$ with $i, j \in \{1, 2, ..., M_H\}$, we have:

$$
\begin{aligned}
&\mathbf{E}_{q(\mathbf{h}_d)} \left[ k^H(\mathbf{h}_d, \mathbf{z}_i^H) \right] \\
&= c \int \mathcal{N}(\mathbf{h}_d \mid \mathbf{z}_i^H, \mathrm{Diag}(l)) \mathcal{N}(\mathbf{h}_d \mid \mathbf{M}_d, \mathbf{\Sigma}_d) d\mathbf{h}_d \\
&= c \, \mathcal{N}(\mathbf{z}_i^H \mid \mathbf{M}_d, \mathbf{\Sigma}_d + \mathrm{Diag}(l)) \int \mathcal{N}\Big(\mathbf{h}_d \mid \underbrace{E(\mathrm{Diag}(l)^{-1}\mathbf{z}_i^H + \mathbf{\Sigma}_d^{-1}\mathbf{M}_d)}_{e}, \underbrace{(\mathrm{Diag}(l)^{-1} + \mathbf{M}_d^{-1})^{-1}}_{E}\Big) d\mathbf{h}_d \\
&= c(2\pi)^{-\frac{Q_H}{2}} \prod_{i=1}^{Q_H} (l_i + s_{d,i})^{-\frac{1}{2}} \exp\Big( -\frac{1}{2} \sum_{i'=1}^{Q_H} \frac{(\mathbf{z}_{i,i'}^H - m_{d,i'})^2}{l_{i'} + s_{d,i'}} \Big) \\
&= \sigma_H^2 \prod_{i=1}^{Q_H} (l_i + s_{d,i})^{-\frac{1}{2}} \prod_{i=1}^{Q_H} l_i^{\frac{1}{2}} \exp\Big( -\frac{1}{2} \sum_{i'=1}^{Q_H} \frac{(\mathbf{z}_{i,i'}^H - m_{d,i'})^2}{l_{i'} + s_{d,i'}} \Big).
\end{aligned}
$$

(19)

Notice that the second line uses the conclusion of the product of Gaussians, which is:

$$
\mathcal{N}(x \mid a, A)\mathcal{N}(x \mid b, B) = z\mathcal{N}(x \mid e, E),
$$

(20)

where $z = \mathcal{N}(a \mid b, A + B)$, and $e = E(A^{-1}a + B^{-1}b)$, $E = (A^{-1} + B^{-1})^{-1}$.

$$
\begin{aligned}
\mathbf{E}_{q(\mathbf{h}_d)} & \left[ k^H(\mathbf{h}_d, \mathbf{z}_i^H) k^H(\mathbf{h}_d, \mathbf{z}_j^H) \right] \\
&= c^2 \int \mathcal{N}(\mathbf{h}_d \mid \mathbf{z}_i^H, \mathrm{Diag}(l)) \mathcal{N}(\mathbf{h}_d \mid \mathbf{z}_j^H, \mathrm{Diag}(l)) \mathcal{N}(\mathbf{h}_d \mid \mathbf{M}_d, \boldsymbol{\Sigma}_d) d\mathbf{h}_d \\
&= c^2 \mathcal{N}(\mathbf{z}_i^H \mid \mathbf{z}_j^H, 2\mathrm{Diag}(l)) \int \mathcal{N}\left( \mathbf{h}_d \mid \frac{\mathbf{z}_i^H + \mathbf{z}_j^H}{2}, \frac{\mathrm{Diag}(l)}{2} \right) \mathcal{N}(\mathbf{h}_d \mid \mathbf{M}_d, \boldsymbol{\Sigma}_d) d\mathbf{h}_d \\
&= c^2 \mathcal{N}(\mathbf{z}_i^H \mid \mathbf{z}_j^H, 2\mathrm{Diag}(l)) \, \mathcal{N}\left( \mathbf{M}_d \mid \frac{\mathbf{z}_i^H + \mathbf{z}_j^H}{2}, \frac{\mathrm{Diag}(l)}{2} + \boldsymbol{\Sigma}_d \right) \\
&= \sigma_H^4 \prod_{i=1}^{Q_H} \left( \frac{l_i}{2} \right)^{\frac{1}{2}} \left( \frac{l_i}{2} + s_{d,i} \right)^{-\frac{1}{2}} \exp\left\{ -\frac{1}{2} \sum_{i'=1}^{Q_H} \left[ \frac{(\mathbf{z}_{i,i'}^H - \mathbf{z}_{j,i'}^H)^2}{2l_{i'}} + \frac{(m_{d,i'} - \frac{\mathbf{z}_{i,i'}^H + \mathbf{z}_{j,i'}^H}{2})^2}{s_{d,i'} + \frac{l_i}{2}} \right] \right\},
\end{aligned}
\tag{21}
$$

by using formulae Eq. (19) and Eq. (21), both statistics $\Psi_d^H$ and $\Phi_d^H$ can be computed. Therefore, $\psi, \Psi, \Phi$ can also be analytically solved.

## A.2 Parametrisation technique of $q(\mathbf{u})$ in GS-LVMOGP

When $Q = 1$, employing the parameterisation $\mathbf{u} = \mathbf{L}\mathbf{u}_0$, as detailed in Section 3.2, results in the covariance matrix of $q(\mathbf{u})$ also exhibiting a Kronecker product structure. This is because:

$$
\mathbf{K} = \mathbf{L}\mathbf{L}^\top = \mathbf{K}^H \otimes \mathbf{K}^X = \left( \mathbf{L}_H \mathbf{L}_H^\top \right) \otimes \left( \mathbf{L}_X \mathbf{L}_X^\top \right) = \left( \mathbf{L}_H \otimes \mathbf{L}_X \right) \left( \mathbf{L}_H^\top \otimes \mathbf{L}_X^\top \right),
\tag{22}
$$

thus, $\mathbf{L} = \mathbf{L}_H \otimes \mathbf{L}_X$ and,

$$
q(\mathbf{u}) = \mathcal{N}(\mathbf{u} \mid \mathbf{M}_\mathbf{u}, \boldsymbol{\Sigma}_\mathbf{u}) = \mathcal{N}(\mathbf{u} \mid \mathbf{L}\mathbf{M}_0, \mathbf{L}(\boldsymbol{\Sigma}_0^H \otimes \boldsymbol{\Sigma}_0^X)\mathbf{L}^\top) = \mathcal{N}(\mathbf{u} \mid \mathbf{L}\mathbf{M}_0, (\mathbf{L}_H \boldsymbol{\Sigma}_0^H \mathbf{L}_H^\top) \otimes (\mathbf{L}_X \boldsymbol{\Sigma}_0^X \mathbf{L}_X^\top)).
\tag{23}
$$

For $Q > 1$, the Cholesky factor $\mathbf{L}$ can not be factorised in general, so the covariance matrix of $q(\mathbf{u})$ does not have Kronecker product structure.

Another advantage of the proposed parametrisation technique is that we have more efficient computation of the $\mathrm{KL}(q(\mathbf{u})\|p(\mathbf{u}))$ term in the ELBO. Firstly, notice

$$
\mathrm{KL}(q(\mathbf{u})\|p(\mathbf{u})) = \mathrm{KL}(q(\mathbf{u}_0)\|p(\mathbf{u}_0)),
\tag{24}
$$

for two general Gaussian distributions $q(\mathbf{u})$ and $p(\mathbf{u})$, computation of $\mathrm{KL}(q(\mathbf{u})\|p(\mathbf{u}))$ is based on the following formula, with $\mathcal{O}(M_X^3 M_H^3)$ complexity:

$$
\mathrm{KL}(q(\mathbf{u})\|p(\mathbf{u})) = \frac{1}{2} \left( \mathrm{Tr}(\mathbf{K}_{\mathbf{u},\mathbf{u}}^{-1} \boldsymbol{\Sigma}_\mathbf{u}) - M_H M_X + \mathbf{M}_\mathbf{u}^\top \mathbf{K}_{\mathbf{u},\mathbf{u}}^{-1} \mathbf{M}_\mathbf{u} + \log\left( \frac{\det \mathbf{K}_{\mathbf{u},\mathbf{u}}}{\det \boldsymbol{\Sigma}_\mathbf{u}} \right) \right),
\tag{25}
$$

while the KL divergence between a Kronecker product structured Gaussian distribution and a standard Gaussian distribution, the $\mathrm{KL}(q(\mathbf{u}_0)\|p(\mathbf{u}_0))$ can be largely simplified, with only complexity $\mathcal{O}(M_X^3 + M_H^3)$:

$$
\begin{aligned}
\mathrm{KL}(q(\mathbf{u}_0)\|p(\mathbf{u}_0)) &= \frac{1}{2} \left( \mathrm{Tr}(\boldsymbol{\Sigma}_0^H \otimes \boldsymbol{\Sigma}_0^X) - M_H M_X + \mathbf{M}_0^\top \mathbf{M}_0 + \log\left( \frac{\det \mathbb{I}}{\det \boldsymbol{\Sigma}_0^H \otimes \boldsymbol{\Sigma}_0^X} \right) \right) \\
&= \frac{1}{2} \left( \mathrm{Tr}(\boldsymbol{\Sigma}_0^H) \mathrm{Tr}(\boldsymbol{\Sigma}_0^X) - M_H M_X + \mathrm{Tr}(\mathbf{M}_0^\top \mathbf{M}_0) - M_H \log\det \boldsymbol{\Sigma}_0^X - M_X \log\det \boldsymbol{\Sigma}_0^H \right).
\end{aligned}
\tag{26}
$$

## A.3 Gauss-Hermite quadrature

Gauss-Hermite quadrature is a numerical technique specifically designed for computing integrals of functions that have a Gaussian (or exponential) weight function. This method is particularly useful when dealing with integrals where the integrand involves a Gaussian-weighted function, making it an ideal choice for expectations involving Gaussian distributions in probabilistic modelling, and in particular, Gaussian process models.

In Gauss-Hermite, the weight function is $e^{-x^2}$, the integral it approximates has form $\int e^{-x^2} g(x)dx$. The approximation is given by:

$$\int_{-\infty}^{+\infty} e^{-x^2} g(x)dx \approx \sum_i w_i g(x_i), \tag{27}$$

where $x_i$ are the roots of the n-th Hermite polynomial, $H_n(x)$, and $w_i$ are the corresponding weights, calculated as: $w_i = \frac{2^{n-1}n!\sqrt{\pi}}{n^2[H_{n-1}(x_i)]^2}$.

### A.3.1 Numerical integration of $\mathcal{L}_{dn}(\mathbf{H}_d) = \mathbb{E}_{q(f_{dn}|\mathbf{H}_d, \mathbf{x}_n)}\left[\log p(y_{dn} \mid f_{dn})\right]$

Recall $q(f_{dn} \mid \mathbf{H}_d, \mathbf{x}_n)$ are Gaussian distribution denoted as $\mathcal{N}(f_{dn} \mid a, b^2)$, where

$$a = \mathbf{K}_{f_{dn}, \mathbf{u}}\mathbf{K}_{\mathbf{u}, \mathbf{u}}^{-1}\mathbf{M_u}; \quad b = \mathbf{K}_{f_{dn}, f_{dn}} + \mathbf{K}_{f_{dn}, \mathbf{u}}\mathbf{K}_{\mathbf{u}, \mathbf{u}}^{-1}(\mathbf{\Sigma_u} - \mathbf{K_{u,u}})\mathbf{K}_{\mathbf{u}, \mathbf{u}}^{-1}\mathbf{K}_{\mathbf{u}, f_{dn}}. \tag{28}$$

To apply Gaussian-Hermite quadrature, we first re-write the integration $\mathcal{L}_{dn}(\mathbf{H}_d)$ by change of variables:

$$x = \frac{f_{dn} - a}{\sqrt{2}b}; \quad df_{dn} = \sqrt{2}bdx \tag{29}$$

This transforms the integral to

$$\int_{-\infty}^{+\infty} e^{-x^2} \frac{1}{\sqrt{\pi}} \log p(y_{dn} \mid a + \sqrt{2}bx)dx,$$

Now we apply Gauss-Hermite quadrature,

$$\int_{-\infty}^{+\infty} e^{-x^2} \frac{1}{\sqrt{\pi}} \log p(y_{dn} \mid a + \sqrt{2}bx)dx \approx \sum_{i=1}^{n} w_i \frac{1}{\sqrt{\pi}} \log p(y_{dn} \mid a + \sqrt{2}bx_i), \tag{30}$$

### A.3.2 Practical Steps

- Determine the degree $n$: the choice of $n$ balances between computational cost and accuracy. In our experiments, we use $n = 20$.

- Find $x_i$ and $w_i$: typically be looked up in numerical libraries or computed using software that handles numerical analysis.

- Evaluate $\log p(y_{dn} \mid f_{dn})$: Compute this term at $f_{dn} = a + \sqrt{2}bx_i$ for each $i$.

- Compute the weighted sum, which is the approximation of the integral.

### A.4 Extended Related Works

**Connections to Neural Process**

Neural Processes (Garnelo et al., 2018b;a) (NP) are a class of models designed for probabilistic meta-learning. In the typical setup, the NP model is trained on a set of tasks or datasets, enabling it to make more accurate probabilistic predictions for new tasks with limited context data. In the case of MOGP, a connection to NP can be made by interpreting each output as a distinct task and considering the missing values for certain outputs as the target for new tasks.

Despite this conceptual connection, there are several key differences between MOGP and NP:

- **Modelling assumption** In MOGP, the data for all inputs and outputs are jointly modelled as a GP, and predictions for new inputs are made through Bayesian inference. Specifically, the predictive distribution is computed as a conditional distribution. In contrast, NP does not explicitly model the joint distribution. Instead, NP focuses on constructing a predictive distribution by mapping the context data to a summarising vector "z" via an encoder network. This vector "z", along with the embedded target inputs, is then passed through a decoder to produce the predictive mean and variance.

- **Information transfer and handling new tasks** In MOGP, information transfer between tasks is achieved by modelling covariances between outputs. This allows the model to share information and make predictions based on the relationships between different outputs. In contrast, NP implicitly shares information by jointly estimating the parameters of the encoder and decoder, and when new tasks come, one expects that the encoder and decoder will "generalise" to context and target data for new tasks. Another distinction between MOGP and NP lies in their ability to handle new tasks. Typical MOGP models are not inherently designed to handle new, unseen tasks after training. On the other hand, NP models are specifically designed to handle new tasks by making predictions from limited context data, enabling them to make predictions for unseen tasks.

### A.5 Integration of latent variables for prediction

The prediction problem for given output $d^*$ and input $\mathbf{x}^*$ involves an integration w.r.t uncertain latent variables $q(\mathbf{H}_{d^*})$, as shown in Eq. 13. This integration is generally intractable, and recall in Section 3.4, we adopt an approach to use means of $q(\mathbf{H}_{d^*})$ to approximate the integral. In this section, we provide an alternative approach: a Gaussian approximation (compute first and second moments) for the predictive distribution. Please be aware that this method is **not** employed in our current experiments, and the approach presented here is an optional addition for future research.

$$q(f^* \mid \mathbf{x}^*) = \int q(f^* \mid \mathbf{H}_{d^*}, \mathbf{x}^*) q(\mathbf{H}_{d^*}) d\mathbf{H}_{d^*} = \int q(f^* \mid \{\mathbf{h}_{d^*,q}\}_{q=1}^{Q}, \mathbf{x}^*) \prod_{q=1}^{Q} q(\mathbf{h}_{d^*,q}) d\mathbf{h}_{d^*,q}, \quad (31)$$

we denote the mean and variance for $p(f^* \mid \mathbf{H}_{d^*}, \mathbf{x}^*)$ as $\lambda(\mathbf{H}_{d^*})$ and $\gamma(\mathbf{H}_{d^*})$, where

$$\lambda(\mathbf{H}_{d^*}) = \mathbf{K}_{f^*,\mathbf{u}} \mathbf{K}_{\mathbf{u},\mathbf{u}}^{-1} \mathbf{M}^{\mathbf{u}}; \quad \gamma(\mathbf{H}_{d^*}) = \mathbf{K}_{f^*,f^*} + \mathbf{K}_{f^*,\mathbf{u}} \mathbf{K}_{\mathbf{u},\mathbf{u}}^{-1} (\boldsymbol{\Sigma}^{\mathbf{u}} - \mathbf{K}_{\mathbf{u},\mathbf{u}}) \mathbf{K}_{\mathbf{u},\mathbf{u}}^{-1} \mathbf{K}_{\mathbf{u},f^*}, \quad (32)$$

We consider the first and second moment for $q(f^* \mid \mathbf{x}^*)$, which we denote as $m$ and $v$ respectively.

$$
\begin{aligned}
m &= \int f^* q(f^* \mid \mathbf{x}^*) df^* = \int \int f^* \mathcal{N}(f^* \mid \lambda(\mathbf{H}_{d^*}), \gamma(\mathbf{H}_{d^*})) df^* q(\mathbf{H}_{d^*}) d\mathbf{H}_{d^*} \\
&= \int \lambda(\mathbf{H}_{d^*}) q(\mathbf{H}_{d^*}) d\mathbf{H}_{d^*} = \mathbb{E}_{q(\mathbf{H}_{d^*})}[\lambda(\mathbf{H}_{d^*})].
\end{aligned}
\quad (33)
$$

$$
\begin{aligned}
v &= \int (f^*)^2 q(f^* \mid \mathbf{x}^*) df^* - m^2 = \int \int (f^*)^2 \mathcal{N}(f^* \mid \lambda(\mathbf{H}_{d^*}), \gamma(\mathbf{H}_{d^*})) df^* q(\mathbf{H}_{d^*}) d\mathbf{H}_{d^*} - m^2 \\
&= \int \{\lambda^2(\mathbf{H}_{d^*}) + \gamma(\mathbf{H}_{d^*})\} q(\mathbf{H}_{d^*}) d\mathbf{H}_{d^*} - m^2 = \mathbb{E}_{q(\mathbf{H}_{d^*})}[\lambda^2(\mathbf{H}_{d^*})] + \mathbb{E}_{q(\mathbf{H}_{d^*})}[\gamma(\mathbf{H}_{d^*})] - m^2,
\end{aligned}
\quad (34)
$$

there are three terms to compute: $\mathbb{E}_{q(\mathbf{H}_{d^*})}[\lambda(\mathbf{H}_{d^*})]$, $\mathbb{E}_{q(\mathbf{H}_{d^*})}[\lambda^2(\mathbf{H}_{d^*})]$ and $\mathbb{E}_{q(\mathbf{H}_{d^*})}[\gamma(\mathbf{H}_{d^*})]$,

$$\mathbb{E}_{q(\mathbf{H}_{d^*})}[\lambda(\mathbf{H}_{d^*})] = \mathbb{E}_{q(\mathbf{H}_{d^*})}[\mathbf{K}_{f^*,\mathbf{u}} \mathbf{K}_{\mathbf{u},\mathbf{u}}^{-1} \mathbf{M}^{\mathbf{u}}] = \mathbb{E}_{q(\mathbf{H}_{d^*})}[\mathbf{K}_{f^*,\mathbf{u}}] \mathbf{K}_{\mathbf{u},\mathbf{u}}^{-1} \mathbf{M}^{\mathbf{u}}, \quad (35)$$

$$\mathbb{E}_{q(\mathbf{H}_{d^*})}[\lambda^2(\mathbf{H}_{d^*})] = \mathbb{E}_{q(\mathbf{H}_{d^*})}[(\mathbf{M}^{\mathbf{u}})^{\top} \mathbf{K}_{\mathbf{u},\mathbf{u}}^{-1} \mathbf{K}_{\mathbf{u},f^*} \mathbf{K}_{f^*,\mathbf{u}} \mathbf{K}_{\mathbf{u},\mathbf{u}}^{-1} (\mathbf{M}^{\mathbf{u}})] = (\mathbf{M}^{\mathbf{u}})^{\top} \mathbf{K}_{\mathbf{u},\mathbf{u}}^{-1} \mathbb{E}_{q(\mathbf{H}_{d^*})}[\mathbf{K}_{\mathbf{u},f^*} \mathbf{K}_{f^*,\mathbf{u}}] \mathbf{K}_{\mathbf{u},\mathbf{u}}^{-1} \mathbf{M}^{\mathbf{u}}, \quad (36)$$

$$\mathbb{E}_{q(\mathbf{H}_{d^*})}[\gamma(\mathbf{H}_{d^*})]$$

$$= \mathbb{E}_{q(\mathbf{H}_{d^*})}[\mathbf{K}_{f^*,f^*} + \mathbf{K}_{f^*,\mathbf{u}}\mathbf{K}_{\mathbf{u},\mathbf{u}}^{-1}(\mathbf{\Sigma}^{\mathbf{u}} - \mathbf{K}_{\mathbf{u},\mathbf{u}})\mathbf{K}_{\mathbf{u},\mathbf{u}}^{-1}\mathbf{K}_{\mathbf{u},f^*}] = \mathbb{E}_{q(\mathbf{H}_{d^*})}[\mathbf{K}_{f^*,f^*}] + \mathbb{E}_{q(\mathbf{H}_{d^*})}\left[\mathbf{K}_{f^*,\mathbf{u}}\mathbf{K}_{\mathbf{u},\mathbf{u}}^{-1}\mathbf{\Sigma}^{\mathbf{u}}\mathbf{K}_{\mathbf{u},\mathbf{u}}^{-1}\mathbf{K}_{\mathbf{u},f^*} \right.$$

$$\left. - \mathbf{K}_{f^*,\mathbf{u}}\mathbf{K}_{\mathbf{u},\mathbf{u}}^{-1}\mathbf{K}_{\mathbf{u},f^*}\right]$$

$$= \mathbb{E}_{q(\mathbf{H}_{d^*})}[\mathbf{K}_{f^*,f^*}] + \mathrm{Tr}\left(\mathbb{E}_{q(\mathbf{H}_{d^*})}\left[\mathbf{K}_{f^*,\mathbf{u}}\mathbf{K}_{\mathbf{u},\mathbf{u}}^{-1}(\mathbf{\Sigma}^{\mathbf{u}} - \mathbf{K}_{\mathbf{u},\mathbf{u}})\mathbf{K}_{\mathbf{u},\mathbf{u}}^{-1}\mathbf{K}_{\mathbf{u},f^*}\right]\right)$$

$$= \mathbb{E}_{q(\mathbf{H}_{d^*})}[\mathbf{K}_{f^*,f^*}] + \mathrm{Tr}\left(\mathbf{K}_{\mathbf{u},\mathbf{u}}^{-1}(\mathbf{\Sigma}^{\mathbf{u}} - \mathbf{K}_{\mathbf{u},\mathbf{u}})\mathbf{K}_{\mathbf{u},\mathbf{u}}^{-1}\mathbb{E}_{q(\mathbf{H}_{d^*})}\left[\mathbf{K}_{\mathbf{u},f^*}\mathbf{K}_{f^*,\mathbf{u}}\right]\right),$$

$$(37)$$

integration over $q(\mathbf{H}_{d^*})$ appears in three terms: $\mathbb{E}_{q(\mathbf{H}_{d^*})}[\mathbf{K}_{f^*,f^*}]$, $\mathbb{E}_{q(\mathbf{H}_{d^*})}[\mathbf{K}_{f^*,\mathbf{u}}]$ and $\mathbf{E}_{q(\mathbf{H}_{d^*})}[\mathbf{K}_{\mathbf{u},f^*}\mathbf{K}_{f^*,\mathbf{u}}]$. These terms can be further simplified by Kronecker product decomposition. First, consider:

$$\mathbb{E}_{q(\mathbf{H}_{d^*})}[\mathbf{K}_{f^*,f^*}] = \mathbb{E}_{q(\mathbf{H}_{d^*})}\left[\sum_{q=1}^{Q}\mathbf{K}_{f^*,f^*;q}^{H} \otimes \mathbf{K}_{f^*,f^*;q}^{X}\right]$$

$$= \sum_{q=1}^{Q}\mathbb{E}_{q(\mathbf{H}_{d^*})}[\mathbf{K}_{f^*,f^*;q}^{H}] \otimes \mathbf{K}_{f^*,f^*;q}^{X} \qquad (38)$$

$$= \sum_{q=1}^{Q}\mathbb{E}_{q(\mathbf{h}_{d^*,q})}\left[\mathbf{K}_{f^*,f^*;q}^{H}\right] \otimes \mathbf{K}_{f^*,f^*;q}^{\mathbf{X}},$$

and,

$$\mathbb{E}_{q(\mathbf{H}_{d^*})}[\mathbf{K}_{f^*,\mathbf{u}}] = \mathbb{E}_{q(\mathbf{H}_{d^*})}\left[\sum_{q=1}^{Q}\mathbf{K}_{f^*,\mathbf{u};q}^{H} \otimes \mathbf{K}_{f^*,\mathbf{u};q}^{X}\right]$$

$$= \sum_{q=1}^{Q}\mathbb{E}_{q(\mathbf{H}_{d^*})}\left[\mathbf{K}_{f^*,\mathbf{u};q}^{H}\right] \otimes \mathbf{K}_{f^*,\mathbf{u};q}^{X} \qquad (39)$$

$$= \sum_{q=1}^{Q}\mathbb{E}_{q(\mathbf{h}_{d^*,q})}\left[\mathbf{K}_{f^*,\mathbf{u};q}^{H}\right] \otimes \mathbf{K}_{f^*,\mathbf{u};q}^{X}.$$

Then consider:

$$\mathbf{E}_{q(\mathbf{H}_{d^*})}[\mathbf{K}_{\mathbf{u},f^*}\mathbf{K}_{f^*,\mathbf{u}}]$$

$$= \mathbb{E}_{q(\mathbf{H}_{d^*})}\left[\left\{\sum_{q=1}^{Q}\mathbf{K}_{\mathbf{u},f^*;q}^{H} \otimes \mathbf{K}_{\mathbf{u},f^*;q}^{X}\right\}\left\{\sum_{q=1}^{Q}\mathbf{K}_{f^*,\mathbf{u};q}^{H} \otimes \mathbf{K}_{f^*,\mathbf{u};q}^{X}\right\}\right]$$

$$= \mathbb{E}_{q(\mathbf{H}_{d^*})}\left[\sum_{q=1}^{Q}\sum_{q'=1}^{Q}\left(\mathbf{K}_{\mathbf{u},f^*;q}^{H} \otimes \mathbf{K}_{\mathbf{u},f^*;q}^{X}\right)\left(\mathbf{K}_{f^*,\mathbf{u};q'}^{H} \otimes \mathbf{K}_{f^*,\mathbf{u};q'}^{X}\right)\right]$$

$$= \sum_{q=1}^{Q}\sum_{q'=1}^{Q}\mathbb{E}_{q(\mathbf{H}_{d^*})}\left[\left(\mathbf{K}_{\mathbf{u},f^*;q}^{H} \otimes \mathbf{K}_{\mathbf{u},f^*;q}^{X}\right)\left(\mathbf{K}_{f^*,\mathbf{u};q'}^{H} \otimes \mathbf{K}_{f^*,\mathbf{u};q'}^{X}\right)\right] \qquad (40)$$

$$= \sum_{q=1}^{Q}\sum_{q'=1}^{Q}\mathbb{E}_{q(\mathbf{H_{d^*}})}\left[\mathbf{K}_{\mathbf{u},f^*;q}^{H}\mathbf{K}_{f^*,\mathbf{u};q'}^{H} \otimes \mathbf{K}_{\mathbf{u},f^*;q}^{X}\mathbf{K}_{f^*,\mathbf{u};q'}^{X}\right]$$

$$= \sum_{q=1}^{Q}\sum_{q'=1}^{Q}\mathbb{E}_{q(\mathbf{h}_{d^*,q})}\mathbb{E}_{q(\mathbf{h}_{d^*,q'})}\left[\mathbf{K}_{\mathbf{u},f^*;q}^{H}\mathbf{K}_{f^*,\mathbf{u};q}^{H}\right] \otimes \mathbf{K}_{\mathbf{u},f^*;q}^{X}\mathbf{K}_{f^*,\mathbf{u};q}^{X},$$

where

$$\mathbb{E}_{q(\mathbf{h}_{d*,q})}\mathbb{E}_{q(\mathbf{h}_{d*,q'})}\left[\mathbf{K}^H_{\mathbf{u},f*;q}\mathbf{K}^H_{f*,\mathbf{u};q'}\right]$$

$$= \begin{cases} \mathbb{E}_{q(\mathbf{h}_{d*,q})}\left[\mathbf{K}^H_{\mathbf{u},f*;q}\mathbf{K}^H_{f*,\mathbf{u};q}\right] & \text{if } q = q', \\[2mm] \mathbb{E}_{q(\mathbf{h}_{d*,q})}\left[\mathbf{K}^H_{\mathbf{u},f*;q}\right]\mathbb{E}_{q(\mathbf{h}_{d*,q'})}\left[\mathbf{K}^H_{f*,\mathbf{u};q'}\right] & \text{if } q \neq q'. \end{cases} \tag{41}$$

Therefore, the key to compute $m$ and $v$ relies on the computation of following statistics:

$$\psi^H_{d*,q} = \mathbb{E}_{q(\mathbf{h}_{d*,q})}\left[\mathbf{K}^H_{f*,f*;q}\right]; \quad \Psi^H_{d*,q} = \mathbb{E}_{q(\mathbf{h}_{d*,q})}\left[\mathbf{K}^H_{f*,\mathbf{u};q}\right], \quad \Phi^H_{d*,q} = \mathbb{E}_{q(\mathbf{h}_{d*,q})}\left[\mathbf{K}^H_{\mathbf{u},f*;q}\mathbf{K}^H_{f*,\mathbf{u};q}\right]. \tag{42}$$

The computation of the above three statistics is the same as $\psi^H_d, \Psi^H_d, \Phi^H_d$ in Appendix A.1.1.

## A.6 Experiment Settings

We use the following metrics in the experiments: MSE (mean square error), RMSE (root mean square error), SMSE (standardised mean square error), and NLPD (negative log predictive density). $\hat{y}_{dn}$ denotes prediction value and $y_{dn}$ refers to the ground truth:

$$\text{MSE} = \frac{1}{ND}\sum_{n=1}^{N}\sum_{d=1}^{D}(y_{dn} - \hat{y}_{dn})^2, \tag{43}$$

$$\text{RMSE} = \sqrt{\frac{1}{ND}\sum_{n=1}^{N}\sum_{d=1}^{D}(y_{dn} - \hat{y}_{dn})^2}, \tag{44}$$

$$\text{SMSE} = \frac{1}{D}\sum_{d=1}^{D}\frac{\frac{1}{N}\sum_{n=1}^{N}(y_{dn} - \hat{y}_{dn})^2}{\frac{1}{N}\sum_{n=1}^{N}(y_{dn} - \bar{y}_d^{train})^2}, \tag{45}$$

$$\text{NLPD} = \frac{1}{ND}\sum_{n=1}^{N}\sum_{d=1}^{D}\int \log p(y_{dn} \mid f_{dn})q(f_{dn})df_{dn}. \tag{46}$$

With a Gaussian likelihood, we make use of closed-form solutions to the NLPD; otherwise, we approximate it using Gaussian Hermite quadrature.

Some hyperparameters used in experiments are shown in Table 5. Table 6 provides details on the initialisation of the kernel parameters and the mean of the variational distribution of the latent variables. For the Exchange and USHCN experiments, the mean of the variational distribution is initialised using random samples from $\mathcal{N}(0,1)$. For other datasets that involve spatial information for each output, we initialise the mean using the spatial coordinates corresponding to each output. For all experiments, the log variances of $q(\mathbf{H})$ are initialised randomly from a standard normal distribution.

The experiments are run on a MacBook Pro with M3 Max and 36 GB of RAM. Except for spatial transcriptomics experiments, all experiments (for each run) are completed in 30 minutes. Spatial transcriptomics experiments take around 8 hours on a laptop.

### A.6.1 Synthetic dataset

In this study, we introduce Stochastic Variational Inference (SVI) approaches for LV-MOGP models, enabling mini-batch training for both input and output data. This approach substantially diminishes computational complexity. However, the inherent stochasticity may also introduce additional "noise" into the optimisation process. To explore this problem, we design a synthetic multi-output dataset by using a function $f(x) =$

Table 5: $M_H$ refers to the number of inducing points on the latent space, $M_X$ refers to the number of inducing points on the input space. $Q_H$ denotes the dimensionality of the latent space. $J$ is the number of samples used in the Monte Carlo estimation of the integration w.r.t. $q(\mathbf{H}_d)$. lr refers to learning rates. Mini-batch size and the number of iterations are also reported. All experiments use Adam optimiser (Kingma & Ba, 2014).

| | $M_H$ | $M_X$ | $Q_H$ | $J$ | Optimizer | Mini-batch size | Iterations | lr |
|---|---|---|---|---|---|---|---|---|
| Exchange | 20 | 50 | 3 | 3 | Adam | 500 | 5000 | 0.01 |
| USHCN | 10 | 50 | 2 | 1 | Adam | 500 | 10000 | 0.1 |
| Spatio-Temporal | 10 | 20 | 2 | 1 | Adam | 500 | 5000 | 0.1 |
| NYC Crime Count | 20 | 50 | 2 | 1 | Adam | 1000 | 5000 | 0.1 |
| Spatial Transcriptomics | 50 | 50 | 3 | 1 | Adam | 1000 | 200000 | 0.1 |

Table 6: More details about the initialisation of the kernel parameters and the latent variables.

| | Outputscale | $k_X$ lengthscale | $k_H$ lengthscale | $q_H$ mean |
|---|---|---|---|---|
| Exchange | 0.1 | 0.1 | 0.1 | Random sample from standard normal |
| USHCN | 1.0 | 1.0 | 0.01 | Random sample from standard normal |
| Spatio-Temporal | 1.0 | 1.0 | 1.0 | Spatial Coordinates |
| NYC Crime Count | 1.0 | 0.01 | 0.1 | Spatial Coordinates |
| Spatial Transcriptomics | 1.0 | 1.0 | 0.01 | Spatial Coordinates |

$\sin^2(ax + b) + \cos(cx) + dx^3 + ex^2 + fx$. The coefficients are random variables with following distributions: $a \sim \text{Uniform}(2\pi, 3\pi)$; $b \sim \text{Uniform}(-1, 1)$; $c \sim \text{Uniform}(2\pi, 3\pi)$; $d \sim \text{Uniform}(-1, 1)$; $e \sim \text{Uniform}(-1, 1)$; ; $f \sim \text{Uniform}(-1, 1)$. We sample the coefficients 100 times to create a 100-output regression problem. Each output has 100 inputs within $[-1, 1]$, with 50 for training and 50 for testing.

Fig. 4 illustrates the training loss trajectories for varying mini-batch sizes. The figure indicates that (1) our algorithm converges across all mini-batch sizes, notably even for small mini-batches, which exhibit greater stochasticity and are often considered noisier, and (2) smaller mini-batch sizes tend to achieve faster convergence in terms of epoch count, likely due to the more frequent optimisation steps associated with smaller batches. This feature of our model facilitates the practical use of small batch sizes, which are more manageable.

Table 7: Experimental results on a synthetic dataset under varied mini-batch configurations, $\mathcal{B}_O$ and $\mathcal{B}_X$ are explained in Fig. 4

| $\mathcal{B}_O \setminus \mathcal{B}_X$ | 10 | 20 | 50 |
|---|---|---|---|
| 20 | Time: $293.3 \pm 10.7$
SMSE: $0.235 \pm 6.2\text{e-}2$
NLPD: $0.597 \pm 9\text{e-}2$ | Time: $204.3 \pm 5.8$
SMSE: $0.224 \pm 3.2\text{e-}2$
NLPD: $0.576 \pm 4\text{e-}2$ | Time: $235.5 \pm 7.2$
SMSE: $0.217 \pm 3\text{e-}2$
NLPD: $0.563 \pm 5\text{e-}2$ |
| 50 | Time: $297.4 \pm 0.49$
SMSE: $0.218 \pm 7\text{e-}2$
NLPD: $0.570 \pm 2\text{e-}2$ | Time: $239.3 \pm 5.6$
SMSE: $0.216 \pm 4\text{e-}2$
NLPD: $0.555 \pm 4\text{e-}2$ | Time: $297.4 \pm 1.2$
SMSE: $0.218 \pm 7\text{e-}2$
NLPD: $0.569 \pm 2\text{e-}2$ |
| 100 | Time: $238.2 \pm 7.8$
SMSE: $0.212 \pm 4\text{e-}2$
NLPD: $0.545 \pm 8\text{e-}2$ | Time: $262.4 \pm 4.4$
SMSE: $0.235 \pm 3\text{e-}2$
NLPD: $0.609 \pm 8\text{e-}2$ | Time: $397.1 \pm 5.2$
SMSE: $0.233 \pm 8\text{e-}2$
NLPD: $0.632 \pm 2\text{e-}2$ |

Table 7 shows the test performance of models trained with varied mini-batch sizes. Performance variations are evident among models trained with different mini-batch sizes. Notably, models trained with the largest mini-batches do not achieve the highest test performance. In contrast, models trained with moderate batch sizes generally show superior performance (e.g., $B_O = 100, B_X = 10$).

### A.6.2 Exchange dataset

The ten international currencies are the following: CAD, EUR, JPY, GBP, CHF, AUD, HKD, NZD, KRW, MXN, and the three precious metals are gold, silver, and platinum. We make plots for the outputs corresponding to CAD, JPY and AUD, as shown in Fig. 5. We also report the error bars for GS-LVMOGP on the exchange dataset, shown in Table 8.

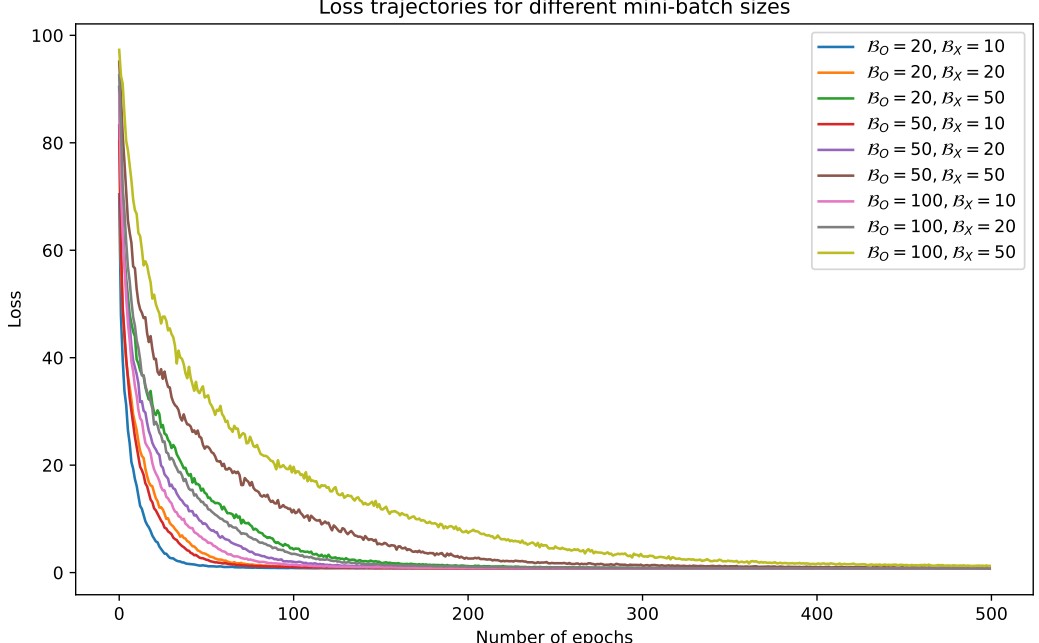

Figure 4: Training loss trajectories on the synthetic dataset with different mini-batch sizes. $\mathcal{B}_O$ represents the output batch size, which indicates the quantity of outputs incorporated within each mini-batch. $\mathcal{B}_X$ denotes the input batch size per output, elucidating the count of training instances selected for every output. The product $\mathcal{B}_O\mathcal{B}_X$ defines the effective size of each mini-batch.

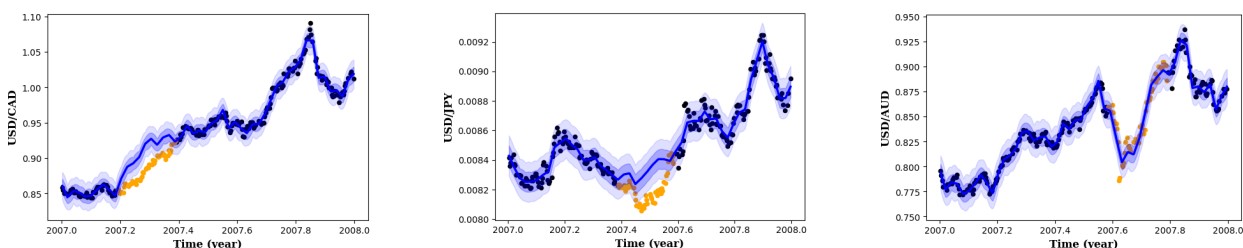

Figure 5: Predictions of the GS-LVMOGP ($Q = 3$) for the exchange rates experiment. Predictions are shown in blue. The shaded area is the predictive mean $\pm$ one and two predictive standard deviations. Training data are denoted as black dots (●) and held-out test data as orange dots (●).

Table 8: The experimental results for GS-LVMOGP on the exchange dataset (with standard deviation).

|  | GS-LVMOGP | | |
|---|---|---|---|
|  | $Q = 1$ | $Q = 2$ | $Q = 3$ |
| SMSE | 0.256±0.1 | 0.186±0.019 | 0.167±0.019 |
| NLPD | -1.851±0.9 | -2.416±0.24 | -2.703±0.19 |

Regarding HetMOGP baseline Moreno-Muñoz et al. (2018), we tried their model for settings with number of latent functions $L = 1, 2, 3, 4, 5$. The results are shown in Table 9. The best results ($L = 2$) are reported in

Table 9: HetMOGP on Exchange datasets with different number of latent functions $L$

|  | $L = 1$ | $L = 2$ | $L = 3$ | $L = 4$ | $L = 5$ |
|---|---|---|---|---|---|
| SMSE | 1.05 | **0.92** | 2.0 | 2.0 | 2.1 |
| NLPD | -1.27 | $\mathbf{-1.3}$ | -1.2 | -1.19 | -1.19 |

Table 1 in the main text.

### A.6.3 Spatiotemporal dataset

The dataset can be downloaded from `https://cds.climate.copernicus.eu/cdsapp#!/dataset/project ions-cmip5-monthly-single-levels?tab=form`, and we only consider the spatial region plotted in Fig. 1, that is, longitude from 13.125°W to 69.375°E, latitude from 33.75°N to 73.75°N. The temperature is measured in Kelvin units.

For the OILMM (Bruinsma et al., 2020) method, there is a hyperparameter $m$, that determines the number of latent processes in the model. We tried a different setting of $m$ from 1 to 100, with preliminary experiment results shown in Table 10, and then we chose $m = 10$ to report results on the main table in Table 3. In the paper, we report the SMSE metric for extrapolated outputs with no training data. The computation of SMSE for them is no longer standardised w.r.t. mean of $y^{train}$ but the mean of $y^{test}$.

In complement with Fig. 2, more output predictions are shown in Fig. 7 and Fig. 8.

Table 10: OILMM model with different number of latent processes ($m$) on Spatio-Temporal dataset.

|  |  | $m = 1$ | $m = 5$ | $m = 10$ | $m = 20$ | $m = 50$ | $m = 100$ |
|---|---|---|---|---|---|---|---|
| imputation | SMSE | 0.276 | 0.252 | 0.134 | 0.130 | 0.788 | 1.0 |
|  | NLPD | 2.943 | 2.944 | 2.819 | 36.609 | 19.202 | 12.357 |
| extrapolation | SMSE | 0.285 | 0.258 | 0.155 | 0.202 | 0.798 | 1.0 |
|  | NLPD | 3.045 | 3.049 | 3.063 | 11.022 | 17.778 | 13.127 |

### A.6.4 NYC Crime dataset

Similar to the spatiotemporal dataset, we do preliminary experiments for OILMM with different numbers of latent processes $m$ on the NYC Crime dataset. The results are shown in Table 11, and we again choose $m = 10$ for the main experiments in Table 2 in the paper.

Table 11: OILMM model with different number of latent processes ($m$)

|  | $m = 1$ | $m = 5$ | $m = 10$ | $m = 20$ | $m = 50$ | $m = 100$ |
|---|---|---|---|---|---|---|
| RMSE | 1.856 | 1.857 | 1.857 | 1.858 | 1.858 | 1.865 |
| NLPD | 1.545 | 1.493 | 1.493 | 1.493 | 1.500 | 1.510 |

Similar to spatiotemporal experiments, we can also encode spatial information into the prior distribution of the latent variables $q(\mathbf{H})$. We run an experiment with this setting and compare it against methods in (Hamelijnck et al., 2021), shown in Table 12.

### A.6.5 USHCN-Daily dataset

The United States Historical Climatology Network (USHCN) dataset, as detailed by Menne et al. (2015), includes records of five climate variables across a span of over 150 years at $1,218$ meteorological stations throughout the United States. It is publicly available and can be downloaded at the following address: `https://cdiac.ess-dive.lbl.gov/ftp/ushcn_daily/`. All state files contain daily measurements for 5 variables: precipitation, snowfall, snow depth, minimum temperature, and maximum temperature.

Table 12: * denotes results taken from (Hamelijnck et al., 2021). Results are averages over 5 cross-validations, and mean and standard deviation are reported. The brackets (G) and (P) represent Gaussian and Poisson likelihoods, respectively.

|  |  | RMSE | NLPD |
|---|---|---|---|
| ST-SVGP* (P) |  | 2.77±0.06 | 1.66±0.02 |
| MF-ST-SVGP* (P) |  | 2.75±0.04 | 1.63±0.02 |
| SVGP-1500* (P) |  | 3.20±0.14 | 1.82±0.05 |
| SVGP-3000* (P) |  | 3.02±0.18 | 1.76±0.05 |
| GS-LVMOGP (G) | $Q=1$ | 2.157±0.062 | 1.500±0.329 |
|  | $Q=2$ | 2.086±0.063 | 1.357±0.132 |
|  | $Q=3$ | 2.075±0.048 | 1.363±0.187 |
| GS-LVMOGP (P) | $Q=1$ | 1.791±0.023 | 1.287±0.003 |
|  | $Q=2$ | 1.791±0.023 | 1.287±0.003 |
|  | $Q=3$ | 1.790±0.024 | 1.287±0.003 |

We follow the same data pre-processing procedure as in De Brouwer et al. (2019), and processed data is kindly provided in `https://github.com/edebrouwer/gru_ode_bayes`. We further filter out outputs with 2 or fewer training observations. The final dataset in the experiment has $5,507$ outputs and $289,144$ training data points.

We also do preliminary experiments for OILMM with different numbers of latent processes $m$ on the USHCN dataset. Based on the results shown in Table 13, we again choose $m = 10$ for the main experiments in Table 4.

Table 13: OILMM model with different number of latent processes ($m$)

|  | $m=1$ | $m=5$ | $m=10$ | $m=20$ | $m=50$ | $m=100$ |
|---|---|---|---|---|---|---|
| MSE | 0.894 | 0.894 | 0.893 | 0.894 | 0.894 | 0.894 |
| NLPD | 811.39 | 810.82 | 810.83 | 811.37 | 816.45 | 822.73 |

### A.6.6 Spatial Transcriptomics dataset

In Fig. (6) we show the mean gene expression of the remaining 4 clusters, where, similarly to cluster 3 (3c), clusters 0 (6a) and 1 (6b) align with the normal tissue area. Cluster 2 (6c) and cluster 4 (6d) contain a combination of normal and tumour regions. It is good to note here that the $5,000$ highly variable genes used in the model are mainly expressed in the normal area of the tissue, which makes that area more prevalent in the clusters. In future applications, a larger gene sample with genes that are enriched in other areas of the tissue would better represent its heterogeneity, and therefore, the clusters will be better resolved.

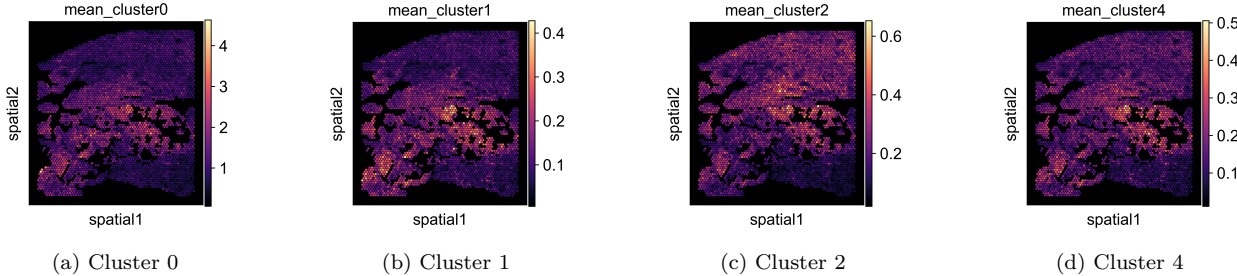

(a) Cluster 0    (b) Cluster 1    (c) Cluster 2    (d) Cluster 4

Figure 6: Latent variable *k-means* clustering results of the prostate carcinoma dataset.

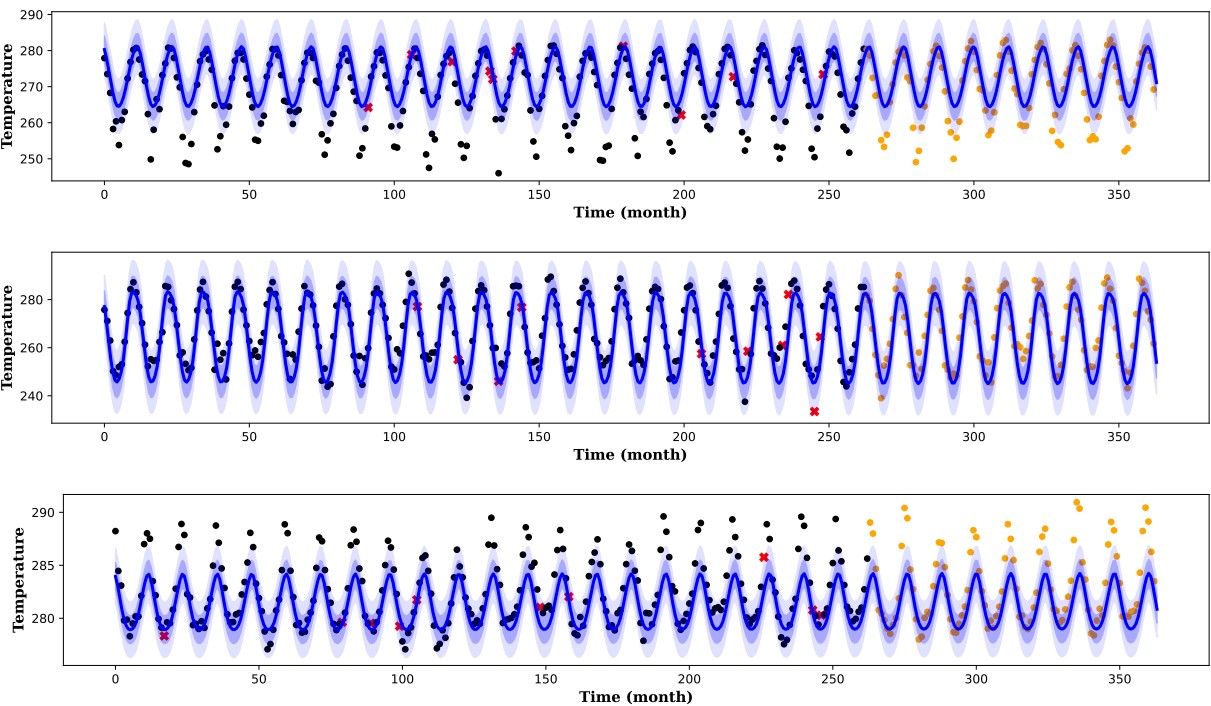

Figure 7: From top to bottom, prediction plots for locations at F, G and H as marked in Fig. 1. Training points are indicated by red crosses (✗), imputation test data by black dots (●), and extrapolation test points by orange dots (●).

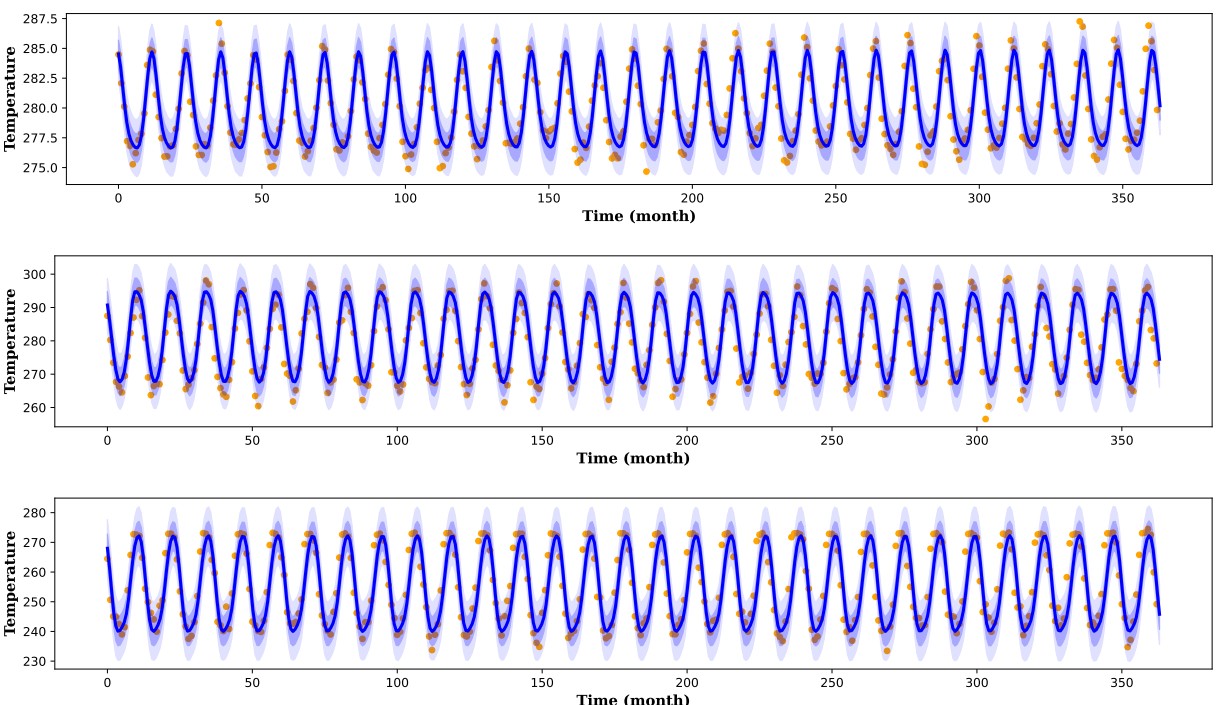

Figure 8: From top to bottom, prediction plots for locations at B, D, and E as marked in Fig. 1. The extrapolation test points are denoted by orange dots (●).

