# OpenReview forum: "Scalable Multi-Output Gaussian Processes with Stochastic Variational Inference"
_TMLR — Accepted by TMLR_

### Review · Reviewer_UKUA · 2025-03-27

**Summary Of Contributions:**

Their method combines the Generalised Gaussian Process Latent Variable Model (Generalized GPLVM) [1], the Latent Variable Multi Output Gaussian Process (LVMOGP) [2], and the Linear Model of Coregionalisation (LMC) [3].

The work focuses on the Multi-Output setting where a single set of inputs will be considered and the goal is to learn multiple output functions in a way which shares information between them. The approach to be used in large data settings via minibatching and non-Gaussian likelihoods via numerical integration similar to [1]. The authors combine the approaches from [2] and [3] to share information between the outputs while maintaining computational effeciency.

[1] Vidhi Lalchand, Aditya Ravuri, and Neil D Lawrence. Generalised gaussian process latent variable models (gplvm) with stochastic variational inference. arXiv preprint arXiv:2202.12979, 2022.

[2] Zhenwen Dai, Mauricio Álvarez, and Neil Lawrence. Efficient modeling of latent information in supervised
learning using gaussian processes. Advances in Neural Information Processing Systems, 30, 2017.

[3] Andre G. Journel and Charles J. Huijbregts. Mining Geostatistics. Academic Press, London, 1978. ISBN 0-12391-050-1.

**Audience:**

Yes

**Broader Impact Concerns:**

For this work I would not be concerned about the ethical implications.

**Claims And Evidence:**

Yes

**Requested Changes:**

Main comments:
- Clarification on which approach is used for finding the predictive density. I feel that the authors should stick to one approach for marginalizing out the latent variales for both the ELBO derivation and the predictions when finding the ELBO, or to provide a justification for why they mix so many techniques to solve the same problem. At minimum the authors need to be very clear about which approach was used for the results that they report.
- Provide details on the initialisation of the kernel parameters and the latent variable locations.
- The conclusion needs to be proofread for clarity and errors. While the majority of the paper is clear the conclusion is not. The authors make some erros in the use of parenthasese in their citations in this section. Additionally, while the authors have a very clear explaination of the ICM and LMC in the introduction, in the conclusion this discussion much less clear.
- The authors have an excellent discussion of the existing Gaussian Process literature for multi task learning, however other communities  have also tackeled similar problems. For example, the Neural Process literature also addresses this form of problem and would be applicable in the larger data set sizes focused on by this paper. I would want to see at least a discussion of how this work relates to that literature, even if the authors consider it out of scope for comparison methods of this paper.


Additional comments:
- Code availability. The proposed method could have a wide range of practical applications, however for these methods the challenge is initially building the model. A major contribution of this kind of work is that the code is made available for other people to use easily, I would like to see the code made available for this work.
- While it is a matter of personal preference, I found the use of $Q$ for the number of different kernels, combined with $Q_h$ for the latent dimensionality of the latent space confusing on the first reading. I would have appreciated a distinct notation for this (eg. use Q for the number of addative components)
- Several references are to the Arxiv versions of the paper, despite having been published in PMLR or other venues. Please use the published rather than the pre-print citation.
- Page 3: "if no extra information" -> "if there is no extra information"
- Page 3: "larger number" -> "large number"

**Strengths And Weaknesses:**

The paper is well written, provides clear background, good experimentation to test their method. The paper makes an important contribution to the line of work folloing from LVMOGPs especially, which is a challenging technique to understand and the work presented here will make the technique far more widely accessable than the original paper did through a clear explaination of how the technique can be applied more broadly.

The two main weaknesses of the paper are:
- Insuffecient detail on the practical optimization procedure. LVMOGPs are typically challenging to train and require thoughtful initialization of the latent variables and GP hyper-parameters. As these are typically very important, the ommision of them from the paper weakens it.
- Inconsistent approach to numerical integration, especially for prediction.  This paper requires taking integrals with respect to the normal distribution several times and they use different approximations each time:

    1.  Integrating the likelihood with respect to the GP predictive distribution to find $\mathcal{L}_{dn}(H_d)$, as in equation 10, when a non-gaussian likelihood is used they Gauss-Hermite (GH) quadrature. As they make the mean field assumption on their latent space GH quadrature could be effeciently could also be applied for the latent variables.
    2. Integrating $\mathcal{L}_{dn}(H_d)$ with respect to $q(H_d)$ as in Equation 11. This is done with Monte Carlo (MC) integration.
    3. When making predictions the predictive density $p(f^*|H_{d^*},x)$ the variational parameters. This is done by either using a MAP estimate for the variational posterior or matching the first and second moments as in the original LVMOGP paper.
  The prediction approach is particularly problematic, as it is not clear to me which approach was taken when reporting their results.
    4. In appendix A.1.1. they give the analytic kernel expectations for the SE-ARD kernel, which is the kernel they use in their experiments. These are the challenging terms which you get to avoid calculating by using numerical integration, but are still provided despite not being used.

  The use of so many different techniques to solve the same problem adds additional complexity to a technique which already has several moving parts.


I have made requested changes for both of these issues.

---

> ### Author Response · Authors · 2025-04-17
> **Reply**
>
> We sincerely thank you for your feedback. We provide a detailed response below.
>
> ---
>
> ## 1. Clarification about training and prediction
> We provide a more detailed clarification regarding the approximation methods used for training and prediction, which we hope will address your confusion. During training, the computation of the ELBO involves two types of integration:
>
> * Integration with respect to the predictive distribution to yield $\mathcal{L}_{dn}(\mathbf{H}_d)$.
>
> * Integration with respect to the variational distribution of latent variables to compute $\mathbb{E}[\mathcal{L}_{dn}(\mathbf{H}_d)]$.
>
> The first integration (1) can be performed analytically for Gaussian likelihood, as shown in Equ. (12). For non-Gaussian likelihoods, this is computed using Gaussian-Hermite quadrature. These practices follow the default implementation in GPyTorch "likelihoods."
>
> For the second integration (2), we use Monte Carlo estimation, and the number of samples is set to one in our experiments, which helps accelerate the training process.
>
> For prediction, we simplify by approximating $q(\mathbf{H}_d)$ using its mean to compute the predictive density. Although we present an alternative method using a Gaussian approximation in the Appendix, we acknowledge that this approach is **not used in our experiments** and is provided as a potential addition for future research. We emphasise these points in Section 3.4 and Appendix A.5 of the revised paper.
>
> ---
>
> ## 2. Initialisation details
> More initialisation details are provided in Appendix A.6 in the revised version.
>
> ---
>
> ## 3. Modify the Conclusion section
> We included a discussion of GS-LVMOGP and LMC in the conclusion, which may have caused some confusion. In the revised version, we have simplified the conclusion section by moving the discussion on the connections between LMC and latent variable MOGP to a separate paragraph in Section 2.
>
> ---
>
> ## 4. Discussion of the Connection to Neural Process (NP)
> Neural Processes (NP) are a class of models designed for probabilistic meta-learning. In the typical setup, the NP model is trained on a set of tasks or datasets, enabling it to make more accurate probabilistic predictions for new tasks with limited context data. In the case of MOGP, a connection to NP can be made by interpreting each output as a distinct task and considering the missing values for certain outputs as the target for new tasks.
>
> Despite this conceptual connection, there are several key differences between MOGP and NP:
>
> ### Modelling assumption
> In MOGP, the data for all inputs and outputs are jointly modelled as a GP, and predictions for new inputs are made through Bayesian inference. Specifically, the predictive distribution is computed as a conditional distribution. In contrast, NP does not explicitly model the joint distribution. Instead, NP focuses on constructing a predictive distribution by mapping the context data to a summarising vector "z" via an encoder network. This vector "z", along with the embedded target inputs, is then passed through a decoder to produce the predictive mean and variance.
>
> ### Information transfer and handling new tasks
>
> In MOGP, information transfer between tasks is achieved by modelling covariances between outputs. This allows the model to share information and make predictions based on the relationships between different outputs. In contrast, NP implicitly shares information by jointly estimating the parameters of the encoder and decoder, and when new tasks come, one expects the encoder and decoder will "generalise" to context and target data for new tasks.
>
> Another distinction between MOGP and NP lies in their ability to handle new tasks. Typical MOGP models are not inherently designed to handle new, unseen tasks after training. On the other hand, NP models are specifically designed to handle new tasks by making predictions from limited context data, enabling them to make predictions for unseen tasks.
>
> We included this discussion in Appendix A.4.2 in the revised version of the paper.
>
> ---
>
> ## 5. Typos/additional comments
>
> We appreciate your attention to the typos. Regarding the availability of the code, we have open-source code online, but we can't provide the link here for anonymity. We will provide the link to the code in our paper after the decision.

---

> > ### Comment · Reviewer_UKUA · 2025-04-23
> >
> > Thank you for the detailed response. I still feel that comparing to a Monte Carlo integration for the latnet variable when making the prediction (with more than one sample) would strengthen the paper, but as the approach used in the results is now clear I am generally in favour of acceptance.

---

### Review · Reviewer_z6m6 · 2025-04-03

**Summary Of Contributions:**

The authors introduce a new approach to modelling with multi-output Gaussian processes (MOGPs), which they call generalised scalable latent variable MOGP (GS-LVMOGP). The new approach generalises the latent-variable MOGP (LV-MOGP, Dai et al., 2017) to use multiple latent spaces each with different kernels, allowing for a richer correlation structure among outputs (to me there is some similarity here to the generalisation of dot-product attention to multi-head dot-product attention in transformers). The authors use inducing locations in both input and latent spaces. While the LV-MOGP uses a full-rank Gaussian approximate posterior over the latent variables, the GS-LVMOGP uses diagonal Gaussian approximate posteriors over the latent variables in each space, i.e. they perform MFVI over the latent variables. It seems (from their exposition and the results) that what they lose in richness of latent variable approximate posterior, they more than make up for in their use of multiple latent spaces (perhaps this is because the disentangling of latent variables into independent spaces somehow simplifies their true posteriors?). In addition to a more flexible model, GS-LVMOGP also has the great advantage that the expected-log-likelihood term in its ELBO can be unbiasedly estimated from a minibatch of data, rendering the approach more scalable than LV-MOGP in the number of datapoints and output dimensions. The authors also include details regarding efficient ways to compute the ELBO/KL divergences for the case of Q=1 leveraging the Kronecker structure, but much of this is already in Dai et al., (2017) and so should not be treated as a new contribution. The authors demonstrate impressive performance across a very wide range of real-world settings.

**Audience:**

Yes

**Broader Impact Concerns:**

None.

**Claims And Evidence:**

Yes

**Requested Changes:**

I use this section to describe what I see as weaknesses in more detail, to ask questions, as well as to list typos.

### **Questions**
In table 2, why do the NLPD results for GS-LVMOGP with a Gaussian likelihood *deteriorate* as Q is increased---this seems like the opposite of what one might expect?

### **Weaknesses**
***Exchange Rates Experiment.*** Errorbars are only shown in the appendix and only for the authors' method. They are also quite wide for some cells. Can these be added to the main paper? Bruinsma et al. (2020) and Nguyen et al. (2014) do not include error bars for this experiment in their own papers, so how statistically significant are the results of this experiment in general---what conclusions can readers of this paper safely make? To me it seems that the OILMM (Bruinsma et al., 2020) missing value for NPLD is an important cell as it's possible it would be the best performer for this metric (this baseline has the 3rd best result for the SMSE metric). Since the authors of that paper have made their code so accessible (https://github.com/wesselb/oilmm), and furthermore since the authors of this paper have already run the OILMM in later experiments, would it not be possible to obtain this missing value?

***NYC Crime Experiment.*** Two baselines are considered in this experiment, and both use a Gaussian likelihood. The proposed model is run with a Gaussian likelihood and a Poisson likelihood. With a Gaussian likelihood, the proposed model is the worst performing in terms of RMSE, and only beats *one* of the baselines in terms of NLPD. With a Poisson likelihood, it outperforms all baselines. It seems then that the use of a Poisson likelihood is the reason for strong performance. Is the OILMM method only applicable to Gaussian likelihoods? If so, perhaps the authors could emphasise that the ability to work with non-conjugate likelihoods is a fairly unique advantage of their approach (as demonstrated in this experiment), or if not, then there really should be a row corresponding to OILMM with Poisson likelihood. In any case, I think there should be a row for independent GPs with a Poisson likelihood (i.e. independent SVGPs or something). For example, could the authors lift the best performing baseline that uses a Poisson likelihood in Table 11 into Table 2?

***Computational Complexity.*** LV-MOGP has a computational complexity of $\mathcal{O}\bigl(\text{max}(N, M_H)\text{max}(D, M_X)\text{max}(X, M_H)\bigr)$ (Section 3.1 of Dai et al., 2017), while GS-LVMOGP has a computational complexity of $\mathcal{O}(M_X^3M_H^3)$. So it seems that, while this approach is applicable to larger datasets and more output dimensions, it is *less* scalable than LV-MOGP in the number of inducing points. That said, I am aware that GS-LVMOGP uses $Q$ sets of $M_H$ inducing variables in the latent spaces rather than just one set in one latent space (as in the LV-MOGP). Have I understood this correctly? If so, perhaps you can offer some insight into why this isn't such a problem (or, if it is, emphasise it as a limitation).

### **Typos/Mistakes**
1. To me "evidence" refers to the log marginal likelihood. i.e. ELBO refers to a lower bound of the log marginal likelihood (not of the marginal likelihood). “log-evidence” would be then refer to the log of the log marginal likelihood, which I am sure is not what you intend to refer to in Section 3.2. However, perhaps this is a matter of differing conventions and it is not as well agreed upon as I thought?
2. Section 4, line 7 of first paragraph, "Nguyen et al. (2014)" should all be in parentheses.
3. Table 4, “standard deviations are in brackets”. They are not---I assume this is just a result of copy-pasting Table 3's caption.
4. Line 3 of the first paragraph in the Conclusions section (section 6), “Dai et al. (2017)” should be in parentheses. Same for “Titsias and Lawrence (2010)” in 4th line of second paragraph.
5. Final sentence of Conclusions/Section 6 “explore the imposing structured” should either be “explore the imposing of structured” or “explore imposing structured”.

### **General Comment**
Despite mostly critical comments from me here, I am impressed with the paper and would probably vote for acceptance in its current form if I had to. Though I do think addressing the above would make it even stronger.

**Strengths And Weaknesses:**

### **Strengths**
- Excellent clarity of exposition
- Novel approach with notable scalability and modelling capacity advantages
- Extensive and convincing experimental results

### **Weaknesses**
- Some experimental details that can be improved or clarified (see below)
- Minor typos/mistakes (see below)

---

> ### Author Response · Authors · 2025-04-17
> **Reply**
>
> We sincerely thank you for your feedback. We provide a detailed response below.
>
> ---
>
> ### 1. Computational Complexity
> The GS-LVMOGP introduces $Q$ sets of latent variables, such that for each latent space, there is a corresponding set of $M_H$ inducing variables. This indeed impacts scalability with respect to the number of inducing points when $Q>1$. A more thorough discussion on this topic is provided in Section 3.3 of the revised paper.
>
> ---
>
> ### 2. Experiments and Comments
> We acknowledge that increasing $Q$ can negatively affect model performance in certain settings. This is likely due to the fact that the informative correlations between the outputs are sufficiently captured with smaller values of $Q$, and further increases may lead to "negative transfer" between outputs. This effect resembles overfitting.
>
> The OILMM model is limited to Gaussian likelihood, and we have emphasised this in the "NYC Crime Count Modelling" section, as you suggested.
>
> ---
>
> ### 3. Typos
> We appreciate your attention to the typos. We recognise that the term "evidence" used in the paper could cause confusion. It has been replaced with "marginal likelihood" in the revised version.

---

> > ### Comment · Reviewer_z6m6 · 2025-04-18
> > **Response to Rebuttal**
> >
> > Many thanks for your response and for making many of the changes I suggested. Although I would still like to see a row corresponding to independent SVGPs with Poisson likelihoods in the NYC experiment table, I am most pleased with the revised version of the paper and would be happy to recommend acceptance.

---

> > > ### Author Response · Authors · 2025-04-18
> > > **Thank you**
> > >
> > > Thank you very much for your encouraging feedback. In response to your suggestion, we have included an additional row in the revised version of the NYC experiment table, corresponding to independent SVGPs with Poisson likelihoods. We are truly grateful for the time and effort you have devoted to reviewing our submission.

---

### Review · Reviewer_jbEw · 2025-04-12

**Summary Of Contributions:**

This work generalizes the latent variable MOGP model introduced in [1] by extending it to incorporate multiple latent variables per output, unlike [1], which uses a single latent variable per output. It then derives a variational inference method for this extension to train the model parameters, including kernel hyperparameters, variational inducing inputs, and multiple variational latent variables. Empirically, the proposed method demonstrates improved SMSE and NLPD across various datasets as the number of latent variables increases.

[1] Efficient Modeling of Latent Information in Supervised Learning using Gaussian Processes – NeurIPS 2017

**Audience:**

Yes

**Claims And Evidence:**

Yes

**Requested Changes:**

## Questions

* Why is there a noticeable gap between LV-MOGP and GS-LVMOGP (Q=1) in Table 1? Aren’t they essentially the same method? If they are different, does your method offer better scalability or efficiency compared to LV-MOGP?

* How would the performances of IGP and OILMM be changed if they consider the Poisson likelihood instead of Gaussian? If they rely on Monte Carlo approximation, wouldn't it be feasible ?

* Doesn't increasing the number of learnable variables (variational parameters of latent variable) also increase the risk of overfitting? As shown in Table 1, your method is only good at SMSE, not for NLPD. As shown in Table 2, the NLPD of GS-LVMOGP (G) with Q=2,3 gets worse than that of GS-LVMOGP (G) with Q=1.

## Suggestion
* Could you explain or emphasize more clearly why introducing multiple latent variables is necessary? This part seems less emphasized in current draft.

* Could you  conduct additional large-scale experiments to empirically demonstrate the scalability of your method—for example, by showing reduced training time on large datasets. Otherwise, it may be more appropriate to tone down the claim of “scalability”.

* Could you also investigate the stability of your method in terms of minibatch size ?


## General comment

I feel hard to assess how interesting this work would be to potential readers because the proposed method seems a minor modification of [1].

[1] Efficient Modeling of Latent Information in Supervised Learning using Gaussian Processes – NeurIPS 2017

**Strengths And Weaknesses:**

## Strengths

* This work presents a variational inference method for a new variant of latent variable MOGP.

* The proposed method demonstrates empirical improvements.


## Weaknesses

* The contribution is largely incremental.

* The motivation for introducing multiple latent variables is unclear and not well-justified in the text.

* The scalability of the proposed method is not demonstrated empirically; that is, experiments on large-scale datasets to show training efficiency are absent.

* Sensitivity analysis with respect to the minibatch size ($m_b$), used for improving computational efficiency, is not provided.

---

> ### Author Response · Authors · 2025-04-17
> **Clarification about the Motivation**
>
> We sincerely thank you for your feedback. We provide a detailed response below.
>
> ## 1. Clarification about the Motivation
> The work extends the LV-MOGP from several aspects:
> * The LV-MOGP framework originally derived closed-form ELBO formulas specifically for Gaussian likelihoods, and these formulas differ depending on whether the data contains missing values, complicating implementation. In contrast, we use a factorised ELBO for training, which easily marginalises missing values and accommodates various data settings within a unified framework. Additionally, our method supports mini-batching and can handle potentially non-Gaussian likelihoods, broadening the practical applications of GS-LVMOGP. For example, using a Poisson likelihood is particularly effective for count data, as demonstrated in the NYC crime and Spatial Transcriptomics experiments.
>
> * We replace the separable kernel used in LV-MOGP with the sum of separable kernels ($Q≥1$). This allows for the use of different types of kernels or kernels with varying lengthscales, enabling better capture of complex covariances among outputs. To ensure GS-LVMOGP with $Q>1$ is decomposable not only among inputs but also outputs, we introduce multiple groups of inducing points on the latent spaces.
>
> * By setting $Q>1$, interesting connections can be drawn between GS-LVMOGP and classic LMC by regarding LMC as a special case of GS-LVMOGP with a linear kernel applied for latent variables on the output space. Compared to directly optimising coregionalisation matrix parameters, GS-LVMOGP "variationally optimise" them, preventing them from overfitting. We added a new paragraph about this connection in the background section in the revised version of the paper.

---

> ### Author Response · Authors · 2025-04-17
> **Experiments and Comments**
>
> ## 2. Experiments and Comments
>
> ### Q: The scalability of the proposed method is not demonstrated empirically.
>
> In the first version of the paper, we had already shown the scalability of our method for datasets with over 5000 outputs, such as a spatial transcriptomics dataset. To our best knowledge, no previous MOGP methods have been successfully applied to similar scale problems without restricted assumptions on likelihoods or output structure.
>
> ---
>
> ### Q: Sensitivity analysis with respect to the minibatch size is not provided.
>
> In the first version of the paper, we had included a synthetic dataset experiment in Appendix A.6.1 to examine the trajectory of the training loss with different input and output mini-batch sizes. The experiment demonstrates that the model trained with a smaller mini-batch size converges more quickly in terms of training epochs.
>
> ---
>
> ### Q: IGP and OILMM with Poisson likelihood instead of Gaussian for NYC Crime experiment.
>
> For OILMM, their model heavily relies on the Gaussian likelihood assumption for both learning and inference, which, to the best of our knowledge, cannot be extended to non-Gaussian likelihoods. In the case of IGP, we applied an exact GP to each output. However, since exact GPs require Gaussian likelihoods, this approach cannot be directly extended to Poisson likelihoods. However, to address this concern, we tested the Sparse Variational Gaussian Process (SVGP) equipped with a Poisson likelihood and 1000 inducing variables on the NYC Crime experiment.
>
> |                  | RMSE              | NLPD              |
> |------------------|-------------------|-------------------|
> | SVGP \(P\)         |  3.000 ± 0.035    | 1.939 ± 0.014     |
> | GS-LVMOGP Q=1 \(P\)| **1.791 ± 0.023** | **1.288 ± 0.003** |
>
> The above tables show superior performance of our method with $Q=1$ compared to the independent SVGP with Poisson likelihood.
>
> ---
>
> ### Q: Increasing the number of learnable variables increases the risk of overfitting. As shown in Table 1, your method is only good at SMSE, not for NLPD. As shown in Table 2, the NLPD of GS-LVMOGP (G) with Q=2,3 gets worse than that of GS-LVMOGP (G) with Q=1.
>
> We acknowledge that increasing $Q$ can negatively affect model performance in certain settings. This is likely due to the fact that the informative correlations between the outputs are sufficiently captured with smaller values of $Q$, and further increases may lead to "negative transfer" between outputs. This effect resembles overfitting. However, we argue that in most cases, increasing $Q$ either enhances predictive performance or has a minimal adverse effect on it.
>
> ---
>
> ### Q: Gap between LV-MOGP and GS-LVMOGP (Q=1) in Table 1.
> In Table 1, LV-MOGP is trained without mini-batching (using the entire dataset for each parameter update), and GS-LVMOGP $(Q≥1)$ is normally trained with mini-batching across outputs and inputs. This causes the gap between these two methods.
>
> ---
>
> ### Q: I feel hard to assess how interesting this work would be to potential readers.
>
> We respectfully disagree with this opinion. Our work extends the LV-MOGP model in several significant ways, including enabling mini-batch training, supporting non-Gaussian likelihoods, and providing a flexible design with sums of separable kernels, among other advancements. We present empirical experiments on datasets with a large number of outputs, such as Spatial Transcriptomics. There is a substantial body of work applying scalable GP methods to study Spatial Transcriptomics data, as demonstrated in [1, 2, 3, 4]. MOGPs are well-suited for analysing such datasets, with each gene modelled as an output. However, many previous MOGP methods are limited to a small number of outputs, which has hindered their application to spatial transcriptomics datasets with thousands of outputs. Our work addresses this limitation and opens up new opportunities for applying MOGP methods to Spatial Transcriptomics, a development that we believe will be of significant interest to many potential readers.
>
> [1] SpatialDE: identification of spatially variable genes.
>
> [2] nnSVG for the scalable identification of spatially variable genes using nearest-neighbor Gaussian processes.
>
> [3] Alignment of spatial genomics data using deep Gaussian processes.
>
> [4] Dependency-aware deep generative models for multitasking analysis of spatial omics data.

---

> ### Comment · Reviewer_z6m6 · 2025-04-18
> **Comment on Potential Audience**
>
> In the spirit of constructive discussion, I would like to support the authors' claim that this work would be of interest to TMLR readers.
>
> At a very high level, Gaussian processes are one of the most useful tools available to many practitioners given their inherent interpretability (/reliability/trustworthiness) and the probabilistic nature of their predictions. Extensions to and advances in Gaussian processes for machine learning, such as this work, are therefore likely to see real-world use and deliver real impact. The authors have already outlined the particular novel contributions that their work makes, so I'll refrain from flogging a dead horse on that front.
>
> As a more anecdotal argument I can vouch that, as a TMLR reader, I certainly find the findings of this work to be of interest.

---

> ### Comment · Reviewer_jbEw · 2025-05-10
>
> Thank you to the authors for their responses. I confirm both your replies and the reviews provided by the other reviewers. Regarding the comment on the novelty of the work, I initially believed that the idea from SVGP [1,2], which introduced mini-batch training of variational inducing inputs for Gaussian process, could be directly applied to the training of latent variables in MOGP. My concern was that mini-batch training is already a standard practice in SVGP, so it was unclear whether applying it to the multiple latent variables in MOGP would be of particular interest to TMLR readers. This was not intended to undermine the value of this work, especially considering that it is based on Gaussian processes, regarded a classical method .
>
> However, after considering the authors' explanation that previous MOGP methods for handling a large number of outputs have been less explored, and that this work addresses this problem, it seems that the contribution is indeed of interest to the TMLR community. I will therefore acknowledge the value of this work and lean toward accepting it. It would be much better, however, to clearly state this contribution from this perspective in the revised manuscript.
>
> [1] Gaussian Processes for Big Data - UAI 13
>
> [2] Scalable Variational Gaussian Process Classification - AISTATS 15

---

> > ### Author Response · Authors · 2025-05-11
> > **Add clarification about our contributions**
> >
> > Thank you for your feedback. In response to your suggestion regarding the clarification of our contribution in handling a large number of outputs compared to previous methods, we have included a clarification in the final paragraph of the introduction section in the revised version (highlighted in blue). We greatly appreciate the time and effort you have dedicated to reviewing our submission.

---

### Decision · Action_Editor_R4Sh · 2025-05-19

**Recommendation:** Accept as is

**Comment:**

The reviewers find that the paper provides an interesting and useful addition to the research area of multi-output GPs and convincingly demonstrates the applicability of the proposed method on real-world problems.

**Audience:**

After some internal discussion about the scope and novelty of the paper, the reviewers agree that it would be interesting for at least a part of the TMLR audience. The reviewers also praise the clarity of the presentation.

**Claims And Evidence:**

The reviewers agree that this paper makes an important contribution to the scalability of multi-output GPs and that the experiments support the performance of the method, especially with the additional experiments added during the discussion period.